physical chemistry

antidepressant drug, micellization, adsorption, micellar/interfacial mole fraction, different spectroscopy

**Author for correspondence:**
Dileep Kumar
e-mail: dileepkumar@tdtu.edu.vn

This article has been edited by the Royal Society of Chemistry, including the commissioning, peer review process and editorial aspects up to the point of acceptance.

# Effects of various media on micellization, adsorption and thermodynamic behaviour of imipramine hydrochloride and antimicrobial surfactant mixtures

Yousef G. Alghamdi[1], Malik Abdul Rub[1,2], Dileep Kumar[3,4] and Abdullah M. Asiri[1,2]

[1]Chemistry Department, Faculty of Science, King Abdulaziz University, Jeddah 21589, Saudi Arabia
[2]Center of Excellence for Advanced Materials Research, King Abdulaziz University, Jeddah 21589, Saudi Arabia
[3]Division of Computational Physics, Institute for Computational Science, Ton Duc Thang University, Ho Chi Minh City, Vietnam
[4]Faculty of Applied Sciences, Ton Duc Thang University, Ho Chi Minh City, Vietnam

DK, 0000-0003-2913-5032

The effect of various media (aqueous, NaCl, urea (U) and thiourea (TU)) on the micellization and adsorption activity of varied mixtures of imipramine hydrochloride (IMP) and benzethonium chloride (BZCl) is investigated via tensiometry. In an aqueous medium, the interactions between IMP and BZCl are also evaluated using UV–visible and FTIR spectroscopy. The interaction between components increases with increased mole fraction ($\alpha_1$) of BZCl in the mixed system (IMP + BZCl). Different parameters, such as micellar and the mixed monolayer component composition, the interaction parameters of the solution and the interface, the activity coefficients of the components in solution and at the interface, and thermodynamic parameters, are computed using different proposed theoretical models (i.e. Clint, Motomura, Rubingh and Rosen). The *cmc* values obtained for the pure components and mixtures (IMP + BZCl) of all the compositions are found to be less in NaCl than in the aqueous solution while found more in the presence of U or TU. TU is more effective in increasing the *cmc* of the pure and mixed systems than U. The Gibbs free energy ($\Delta G^{\circ}_{\mathrm{mic}}$) values of the studied pure and mixed systems are negative, showing the spontaneous nature of the reaction.

**Scheme 1.** Molecular model of imipramine hydrochloride (IMP).

# 1. Introduction

Most surface-active agents are amphiphilic organic molecules, consisting of hydrophobic groups in a hydrocarbon tail and hydrophilic groups as the head of a single molecule [1–6]. The association within amphiphilic molecules (such as surfactants) called micelles that happens due to the balance between hydrophilic and hydrophobic interactions is the critical micelle concentration (*cmc*) [7–13]. Because of their exceptional structural characteristics and good solubilization properties [6], micelles have been extensively used in industries such as pharmaceuticals, dyes, make-up and environmental management [4,6]. Recently, the solubilization outcomes of surfactants in micelles have been extensively studied [6]. Micelles are also excellent catalysts [14–18].

In several practical applications, it is acknowledged that two or more types of surfactant mixture are employed in formulations because of their effective solubilization, pharmaceutical and therapeutic formulations, and dispersion along with detergents [19,20]. Compared with the self-association of a single amphiphile, the amphiphile mixtures form mixed associations with excessive efficiency and lower charge. The resulting mixed micelles have much-improved surface activity [19,21,22]. A mixture can have improved air–solution interface properties as well as varied colloidal properties received from both components. Accordingly, in pharmaceutical chemistry, a mixed micelle solution is used to increase the absorption of different medicines in individuals [22,23]. Nevertheless, few studies address the impact of external factors such as electrolytes, urea on the surfactants and drug compounds in micelles. Additional research in this area will significantly ease screening as well as develop micellar systems, realize drug delivery tools and attain drug delivery goals [24,25].

Because it is straightforward to distinguish micelles, *cmc* is typically used to initially assess the solubilizing capability of surfactants [6]. A lower *cmc* value means that a lesser amount of the surfactant is required to form micelles [26]. Consequently, the *cmc* is also important for examining the interactions between drugs and surfactant molecules. Similar to surfactants, several amphiphilic drugs (including tricyclic antidepressants, phenothiazines and tranquillizers) also have the potential to self-associate into micellar forms [27–29]. These amphiphilic drugs are classified based on their different functional groups, for example, hydrophilic and hydrophobic groups, as these groups are possibly the main reason for the therapeutic properties of the drug. Despite their amphiphilic nature, these drugs do not form stable micelles and act as their own carrier because of their low lipophilic nature and low counterion binding [30]. To overcome this situation, a surfactant and amphiphilic drug mixed system is proposed as an alternative to options such as vesicles or soluble polymers. Usually, surfactants consist of a more hydrophobic micellar core that has a superior capability to admit amphiphilic drugs, enhance drug solubility and inhibit drug precipitation [30,31].

In the current study, an amphiphilic drug, imipramine hydrochloride (IMP), which is the hydrochloride salt form of imipramine, is employed as a model drug (scheme 1). This drug is used to treat depression (antidepressants). The IMP molecule comprises a tricyclic ring core with an alkylamine side chain to which a terminal nitrogen atom is attached. Due to the presence of the alkylamine side chain, this drug behaves like a conventional surfactant but forms micelles at a higher concentration than the surfactant [22,32]. The pKa value of IMP has been reported as 9.5; therefore, this drug has a positive charge in the current study condition [20,22].

Apart from the positive outcomes of drugs, they can also have many undesirable side effects. These unwanted effects may be minimized if drugs are properly used with a carrier. The usage of micelles as drug carriers offers many advantages compared with other possible carriers, and nearly all surfactants form micelles [6]. Micelles can solubilize the drugs with low solubility in their hydrophobic interiors and boost bioavailability. Furthermore, micelles are easily prepared on an industrial scale [31,33].

The interaction of benzethonium chloride (BZCl) surfactant (scheme 2) with the drug IMP in aqueous, salt, urea (U) and thiourea (TU) media is studied. BZCl belongs to the category of N-cationic

**Scheme 2.** Structure of benzethonium chloride (BZCl) surfactant.

**Table 1.** Source and purity of the compounds used.

| chemical | source | CAS number | purification methods | mass fraction purity |
|---|---|---|---|---|
| IMP | Sigma (USA) | 113-52-0 | vacuum drying | ≥0.98 |
| BZCl | Sigma (USA) | 121-54-0 | vacuum drying | ≥0.98 |
| NaCl | BDH (England) | 7647-14-5 | vacuum drying | 0.98 |
| U | Sigma (Germany) | 57-13-6 | vacuum drying | 0.98 |
| TU | Techno Pharmachem (India) | 62-56-6 | vacuum drying | 0.995 |

surfactants that have topical anti-infective and antiseptic qualities, which have been acknowledged for an extended period [34]. Because of its ability to form micelles, BZCl can capture hydrophobic compounds in the interior of its hydrophobic core or the Stern layer [35]; thus, it can act as a drug delivery agent by integrating hydrophobic drugs [36]. The IMP drug employed is highly soluble in water but forms micelles at higher concentrations due to low hydrophobicity; therefore, for its safe delivery, a carrier is required. Accordingly, BZCl is employed as a carrier in this study. Solutions containing both components formed mixed micelles, and the *cmc* value of the drug was too greatly reduced; hence, much less of the drug is required for particular purposes. The effects of using NaCl, U and TU media of fixed concentration on IMP and BZCl mixture interactions were also studied since electrolytes and U are present in the body, which may affect the interaction of IMP and BZCl and influence the drug biological activity. The outcomes of electrolyte and ureas can give more knowledge for drug and surfactant mixtures for developing improved delivery systems than aqueous system, since in the presence of electrolyte or ureas, the value of *cmc* of singular and mixture of amphiphiles declines or rises as the spontaneity of solution mixture declines or rises. Detail information is forever valuable in attaining drugs' biological activity having the minimum unwanted effects. The surface tension technique was used to assess the mixed association behaviour of IMP + BZCl mixtures in different ratios, and the experimental outcomes were analysed using four theories of mixed micellization (Clint, Rubingh, Rosen and Motomura). FTIR spectroscopic analysis was also carried out to further confirm the interaction between IMP and BZCl. UV–visible measurements were also taken to understand the interaction mechanism between the constituents employed in this study.

# 2. Material and methods

## 2.1. Materials

Every compound used is of analytic grade and was applied as procured from their corresponding seller with no additional purification (table 1). Distilled water (conductivity approx. 1–6 µS cm$^{-1}$) was used for the solutions preparation in all solvents media.

## 2.2. Methods

### 2.2.1. Surface tension measurement

The surface tension ($\gamma$) measurements were conducted using an Attension tensiometer (Sigma 701, Germany), which applies the platinum ring detachment process at 298.15 K. The $\gamma$-values of stock

solutions of pure and mixed systems (IMP, BZCl and IMP + BZCl mixtures in the presence of various solvents) were determined by adding a fixed quantity of stock solution by micropipette. Similar measurements were repeated until the $\gamma$-value became constant with further addition of the solution. The measured $\gamma$ of pure and mixed systems (IMP + BZCl) in all media versus their respective log concentrations were plotted, and the breakpoint in the plot is the *cmc* for the system. The temperature of the studied system was retained via circulating water from an ORBIT RS10S thermostat connected with the instrument. The error in temperature obtained during the measurement of $\gamma$ was attained ±0.2 K. The relative uncertainties limits on *cmc* were achieved close to 3% and the combined expanded uncertainty of surface tension value was attained close to 1.0 mN m$^{-1}$. The whole evaluated surface tension data of the employed system are provided in electronic supplementary material, tables S1–S4.

### 2.2.2. FTIR spectroscopy

The FTIR spectra of pure IMP, BZCl, and a 1:1 IMP + BZCl mixture in an aqueous solution were measured at 298.15 K using a Nicolet iS50 FTIR spectrometer (Thermo Scientific, Madison, WI, USA). All spectra were accumulated from 4000 to 400 cm$^{-1}$; however, only a portion of this range is displayed for clarity. In all samples, the water background was removed from the spectra.

### 2.2.3. UV–visible spectroscopy

An Evolution 300 UV–visible (Thermo Scientific) spectrometer was used here to record the spectra of IMP solutions with increasing concentration of BZCl in an aqueous solution at room temperature (298.15 K). The UV–visible absorbance spectra were recorded after each increment of BZCl. Distilled water was used for baseline correction.

## 3. Results and discussion

### 3.1. Determination of *cmc* and *cmc*$^{id}$ values

The present research concentrates mainly on the physico-chemical interactions between IMP and the prospective surface-active surfactant BZCl, and several theoretical models are applied to the data collected. The *cmc* value of individual amphiphiles (IMP and BZCl) and binary amphiphile (IMP + BZCl) mixtures in different ratios determined by a tensiometric method. The amphiphile chain length and hydrophobic interaction are the most important qualities for micellization [6]. Consequently, the *cmc* of the amphiphile solution can be determined from the surface tension ($\gamma$) versus concentration plot. The resulting plots of the IMP + BZCl mixtures at 298.15 K in various media (water, 50 mmol kg$^{-1}$ NaCl, 300 mmol kg$^{-1}$ U and 300 mmol kg$^{-1}$ TU) at different mole fractions of BZCl ($\alpha_1$) are shown in figure 1. With the addition of the amphiphile (pure or mixed) solution into the solvent (aqueous or non-aqueous), the surface tension ($\gamma$) value decreased linearly in the pre-micellar region, and after an excess concentration of amphiphile(s) was added to the solvent, the $\gamma$ value obtained became constant. The breakpoint obtained between $\gamma$ and concentration ($C$) is the *cmc* of the amphiphiles (figure 1) [37]. The obtained *cmc* values of the current system are given in table 2 and electronic supplementary material, figure S1. The solubility of IMP in aqueous media was tested above a concentration of 400 mmol kg$^{-1}$, and hence, IMP is found to be freely soluble in an aqueous system.

The *cmc* of IMP is 41.85 mmol kg$^{-1}$ in an aqueous solution at 298.15 K, showing good agreement with earlier published values [20,22,38]. The *cmc* value of BZCl is 2.95 mmol kg$^{-1}$, which is also in good agreement with the literature (table 2) [39,40]. As our best knowledge, the *cmc* value of BZCl by surface tension method was first reported by our group [40]. As shown in table 2, the *cmc* value of BZCl is much lower than the *cmc* of IMP because the surfactant used is much more hydrophobic in nature. Consequently, the BZCl surfactant initiates micelle formation at a lower concentration than IMP.

Table 2 also shows that in the case of a mixed system (IMP + BZCl), the *cmc* of the formed micelles was reduced too much compared with the *cmc* of pure IMP. With the increased molar fraction ($\alpha_1$) of BZCl in the mixtures, the *cmc* decreases more, to below the *cmc* of pure BZCl surfactant, due to the enhanced interactions (table 2 and figure 2). Table 2 also shows that the *cmc* of the mixed system was closer to the *cmc* value of pure BZCl, indicating that the constituent with high hydrophobicity initiates micelle formation at an interior concentration than the less hydrophobic component. Among the components used, BZCl is more hydrophobic than IMP. BZCl forms micelles at a lower concentration and persists at

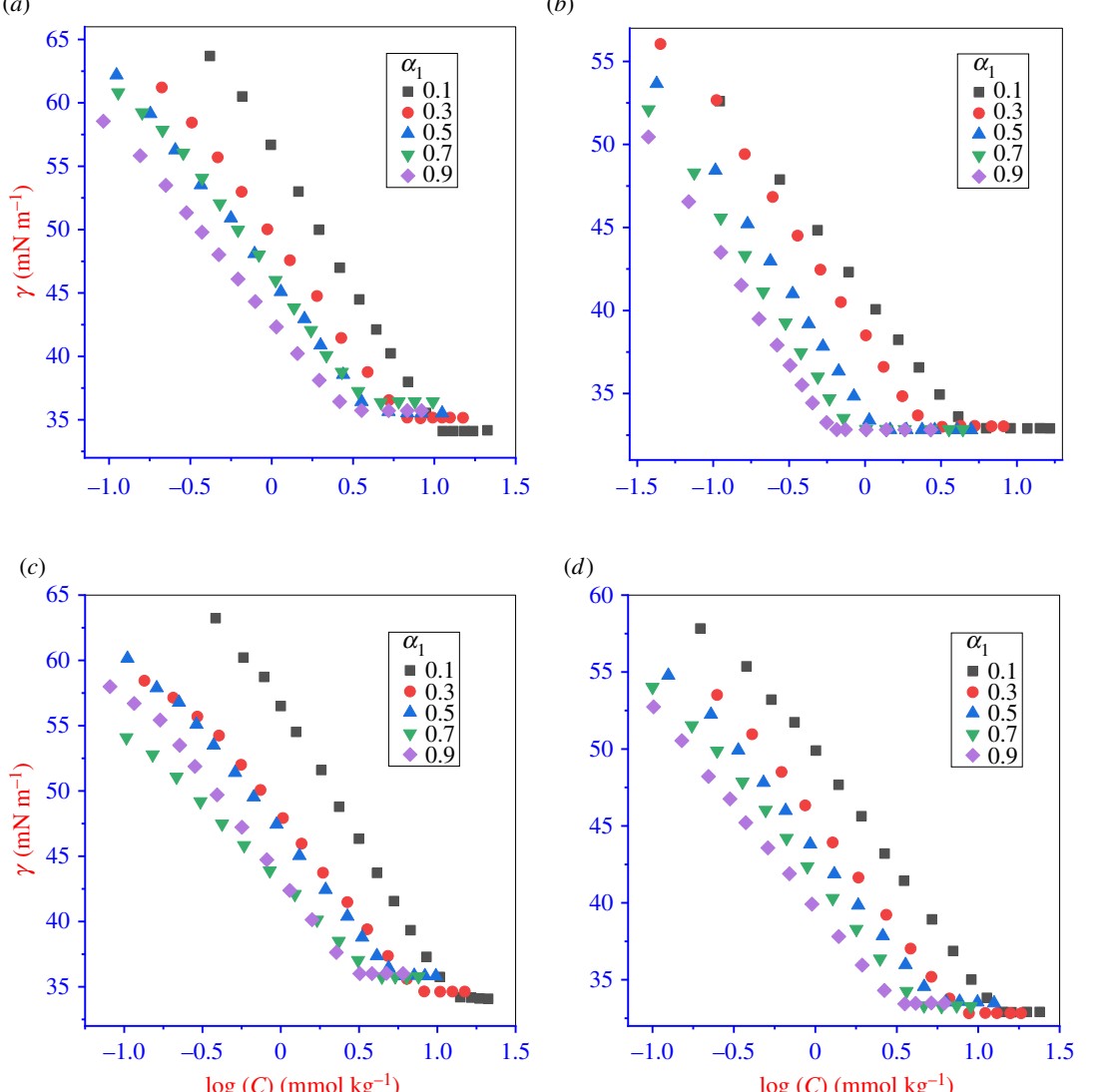

**Figure 1.** Plot of surface tension ($\gamma$) against concentration (C) of IMP + BZCl mixture in five different mole fractions ($\alpha_1$) of BZCl: (a) water, (b) 50 mmol kg$^{-1}$ NaCl, (c) 300 mmol kg$^{-1}$ urea and (d) 300 mmol kg$^{-1}$ thiourea system at 298.15 K.

much higher compositions in mixed micelles. IMP monomers only join the micelles formed with BZCl. Consequently, the mixed micelles which form contain more BZCl than IMP.

The amphiphile mixing can be either ideal or non-ideal throughout mixed micelle formation. The formation of mixed micelles may be represented through the relation [6]. In the case of a binary mixture, the ideal cmc value ($cmc^{id}$) of the mixed micelles was assessed via Clint's theory [41], which is valuable for comparing an ideal and a non-ideal mixed system.

$$\frac{1}{cmc^{id}} = \frac{\alpha_1}{cmc_1} + \frac{\alpha_2}{cmc_2}.$$ 

(3.1)

In equation (3.1), $\alpha_1$, $\alpha_2$, $cmc_1$ and $cmc_2$ signify the mole fraction of BZCl, the mole fraction of IMP, the cmc of BZCl and the cmc of IMP, respectively. Table 2 and figure 2 and visibly show that the $cmc^{id}$ (ideal) in each case is greater than the cmc (the experimental values of the binary mixtures), signifying the existence of non-ideal behaviour because of the mutual interaction of the IMP and BZCl in the mixed system (i.e. attractive interaction or synergistic behaviour). In the IMP + BZCl mixture, the BZCl and the IMP have a high tendency to self-association in solution because of the decreasing cmc of mixtures due to lowered repulsion within the head groups owing to the effective screening effect of BZCl/IMP. The $cmc^{id}$ values are higher than the experimental cmc values because the IMP + BZCl mixture strengthens the hydrophobicity of the mixed system, causing the start of micellization at smaller

**Table 2.** Various physico-chemical parameters for IMP + BZCl mixed systems in various media at 298.15 K.[a]

| $\alpha_1$ | $cmc$ (mmol kg$^{-1}$) | $cmc^{id}$ (mmol kg$^{-1}$) | $X_1^m$ | $X_1^{id}$ | $\beta^m$ | $f_1^m$ | $f_2^m$ | $\ln(cmc_1/cmc_2)$ |
|---|---|---|---|---|---|---|---|---|
| aqueous solution | | | | | | | | |
| 0 | 41.85 | | | | | | | |
| 0.1 | 10.20 | 18.05 | 0.552 | 0.612 | −2.34 | 0.626 | 0.490 | |
| 0.3 | 6.07 | 8.44 | 0.720 | 0.859 | −1.97 | 0.857 | 0.361 | |
| 0.5 | 3.97 | 5.51 | 0.772 | 0.934 | −2.65 | 0.872 | 0.206 | −2.65 |
| 0.7 | 3.60 | 4.09 | 0.879 | 0.971 | −2.0 | 0.971 | 0.214 | |
| 0.9 | 2.72 | 3.25 | 0.878 | 0.992 | −3.80 | 0.945 | 0.053 | |
| 1 | 2.95 | | | | | | | |
| 50 mmol kg$^{-1}$ NaCl | | | | | | | | |
| 0 | 37.25 | | | | | | | |
| 0.1 | 4.56 | 15.66 | 0.535 | 0.622 | −5.03 | 0.338 | 0.237 | |
| 0.3 | 2.38 | 7.26 | 0.622 | 0.864 | −5.52 | 0.455 | 0.118 | |
| 0.5 | 1.15 | 4.72 | 0.637 | 0.937 | −7.79 | 0.358 | 0.043 | −2.69 |
| 0.7 | 0.79 | 3.50 | 0.656 | 0.972 | −9.27 | 0.334 | 0.019 | |
| 0.9 | 0.58 | 2.78 | 0.679 | 0.993 | −11.5 | 0.305 | 0.005 | |
| 1 | 2.52 | | | | | | | |
| 300 mmol kg$^{-1}$ urea | | | | | | | | |
| 0 | 45.20 | | | | | | | |
| 0.1 | 12.05 | 20.13 | 0.549 | 0.599 | −2.10 | 0.653 | 0.532 | |
| 0.3 | 7.20 | 9.54 | 0.728 | 0.852 | −1.69 | 0.883 | 0.408 | |
| 0.5 | 5.15 | 6.26 | 0.815 | 0.931 | −1.79 | 0.940 | 0.304 | −2.60 |
| 0.7 | 4.0 | 4.65 | 0.866 | 0.969 | −2.16 | 0.962 | 0.198 | |
| 0.9 | 2.90 | 3.70 | 0.853 | 0.992 | −4.31 | 0.911 | 0.044 | |
| 1 | 3.36 | | | | | | | |
| 300 mmol kg$^{-1}$ thiourea | | | | | | | | |
| 0 | 46.0 | | | | | | | |
| 0.1 | 12.5 | 20.70 | 0.547 | 0.595 | −2.06 | 0.656 | 0.540 | |
| 0.3 | 7.52 | 9.86 | 0.730 | 0.850 | −1.63 | 0.888 | 0.421 | |
| 0.5 | 5.35 | 6.47 | 0.816 | 0.929 | −1.74 | 0.943 | 0.315 | −2.58 |
| 0.7 | 4.15 | 4.82 | 0.867 | 0.969 | −2.13 | 0.963 | 0.202 | |
| 0.9 | 3.01 | 3.83 | 0.853 | 0.992 | −4.28 | 0.912 | 0.044 | |
| 1 | 3.48 | | | | | | | |

[a]Standard uncertainties ($u$) are $u(T) = 0.20$ K, $u(NaCl) = 1$ mmol kg$^{-1}$, $u(urea) = 2$ mmol kg$^{-1}$, $u(thiourea) = 2$ mmol kg$^{-1}$ and $u(p) = 5$ kPa (level of confidence = 0.68). Relative standard uncertainties ($u_r$) are $u_r(cmc/cmc^{id}) = \pm 3\%$, $u_r(X_1^m/X_1^{id}) = \pm 3\%$, $u_r(\beta^m) = \pm 3\%$ and $u_r(f_1^m/f_2^m) = \pm 4\%$.

concentrations compared with pure species. Consequently, owing to the attractive interaction between the components, $cmc^{id}$ values are higher than the experimental $cmc$ values. Throughout the interaction, electrostatic interactions occur between the micellar head groups, accompanied by the chain–chain interactions between micelles of different chain lengths [42].

Table 2 also shows that in NaCl media, the $cmc$ values of the pure and mixed systems of various ratios were lower than the corresponding aqueous solution. Salt promotes micelle formation by lowering the repulsion between the head groups and thus lessening the effective area captured with every head group [6]. Adding NaCl causes the reduction of the $cmc$ of the pure and mixed systems since a

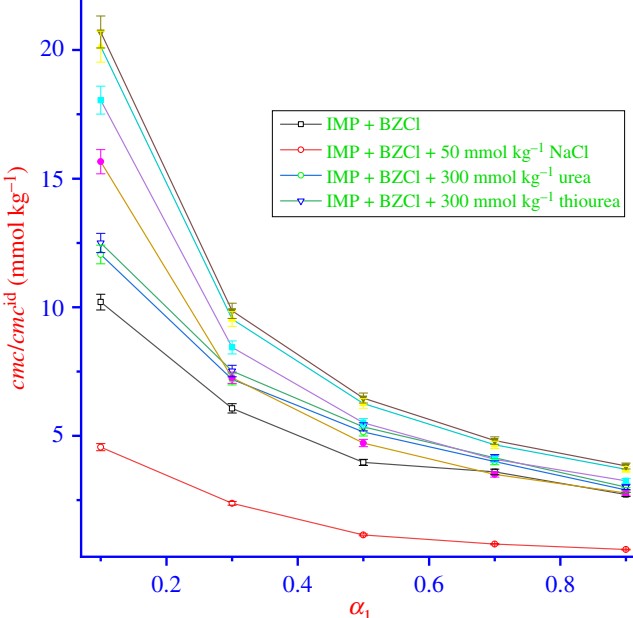

**Figure 2.** Variation of *cmc* and *cmc*<sup>id</sup> value of IMP + BZCl mixtures with $\alpha_1$ variation of BZCl (experimental *cmc* (open symbol) and *cmc*<sup>id</sup> (filled symbol) (relative standard uncertainties ($u_r$) is $u_r(cmc/cmc^{id}) = \pm 3\%$).

decline in electrostatic interactions occurs in NaCl media, and consequently, interactions between the molecules are enhanced, which triggers aggregation at lower concentrations [43].

However, in U or TU media, the *cmc* of the pure and mixed component systems were found to correspond more to those of the aqueous system (table 2), but TU is more efficient than U in increasing the *cmc* values. In the presence of U or TU, the polarity of the medium decreased [44]. The effects of U and TU on the aggregation behaviour of amphiphiles are still poorly understood, but plausible mechanisms are available in the literature for urea [45]. The first is the indirect mechanism, in which U or TU alters only the solvent, resulting in a change in the water structure [46]. The second is the direct mechanism, in which U or TU is involved in hydrophobic solvation by substituting for water particles in the hydration shells [47]. For U or TU, the indirect mechanism is broadly acknowledged [48]; therefore, U and TU act as water-structure breakers [49]. Through the interruption of the water structure by U or TU, the hydration of the polar head groups increases, raising the solubility of unassociated molecules. The result of breaking the water structure is similar to that of increased temperature [50]. TU is a better water-structure breaker than U [51]; therefore, *cmc* rises more when TU is added than when U is added. Additionally, in a solution containing U or TU, the repulsive interactions among the head groups of monomers at the micellar interface are enhanced. Therefore, the aggregation of pure IMP, BZCl and their mixtures is more hindered to some extent than in the aqueous system, and the resulting *cmc* values are higher. In table 2, IMP + BZCl mixtures in NaCl media showed higher non-ideality than in the aqueous, U and TU systems. This non-ideal behaviour measured in the IMP + BZCl mixed system in different media validates the effectiveness of BZCl in strengthening the hydrophobic atmosphere in the mixed state that results in stable mixed micelles.

## 3.2. Mixed micellization study of IMP and BZCl mixed system

From our measurements of *cmc*, BZCl forms mixed micelles with IMP. The quantitative evaluation of non-ideality and the type of interactions between the constituents are understood using Rubingh's theory [52]. Furthermore, the micellar composition (depending on regular solution theory) have been analysed by solving equation (3.2)

$$\frac{(X_1^m)^2 \ln[(\alpha_1 cmc/X_1^m cmc_1)]}{(1 - X_1^m)^2 \ln[(1 - \alpha_1)cmc/(1 - X_1^m)cmc_2]} = 1, \tag{3.2}$$

where $X_1^m$ is the composition of the first constituent (BZCl) in the mixed micelles.

The composition of the first constituent (BZCl) in the mixed micelles in the ideal state $(X_1^{id})$, is computed using equation (3.3) from Motomura & Aratono [53].

$$X_1^{id} = \frac{\alpha_1 cmc_2}{\alpha_1 cmc_2 + \alpha_2 cmc_1}. \tag{3.3}$$

The values of $X_1^m$ of mixed micelles determined at various bulk mole fractions $(\alpha_1)$ of BZCl are given in table 2 in each medium. The $X_1^m$ of BZCl in the mixed micelles is very high compared with $X_2^m$ (the equivalent value for IMP), signalling the greater extent of transfer of BZCl from the solution into the micellar form compared with IMP in all the media studied. The greater occurrence of BZCl in mixed micelles is attributed to the high hydrophobicity of BZCl, greater than IMP. Furthermore, due to the presence of BZCl in the mixed micelles, the electrostatic repulsions between the head groups of IMP declined. The $X_1^m$ values were also larger than the $\alpha_1$ of BZCl used except at the highest $\alpha_1$, reflecting the higher amount of surfactant in the mixed micelles. The $X_1^m$ values in all cases were lower than the $X_1^{id}$ value at each $\alpha_1$ in aqueous, salt, U and TU media, demonstrating that the composition of BZCl in mixed micelles is less than predicted from the ideal values (table 2) [54]. The calculated $X_1^{id}$ value is higher than the $\alpha_1$ value in all corresponding cases, meaning that the calculated value will certainly be higher than $X_1^m$. In addition, $X_1^{id}$ increases because of an increase in $\alpha_1$ of the mixtures (table 2).

Rubingh's theory [52] is also employed to assess the nature and strength of the interactions of the constituents via the interaction parameter $(\beta^m)$ determined by equation (3.4)

$$\beta^m = \frac{\ln(cmc\alpha_1/cmc_1 X_1^m)}{(1 - X_1^m)^2} \tag{3.4}$$

The calculated values of $\beta^m$ in each medium are presented in table 2. As stated in the literature [6], for the mixed system, a positive $\beta^m$ indicates antagonistic behaviour, a negative $\beta^m$ indicates attractive interactions or synergistic effects, and a $\beta^m$ equal to 0 represents ideal behaviour (neither interaction nor repulsion) between both components during mixed micelle formation. In our case, table 2 shows that in all media, the $\beta^m$ values were negative, and their values do not follow any trend with increased $\alpha_1$. Overall, the $\beta^m$ values increased at higher $\alpha_1$, revealing that attractive interactions were detected between the components in the mixed systems. Here, negative $\beta^m$ values were obtained because of the decline in electrostatic self-repulsion of the cationic molecules [55]. Table 2 shows that in all media, the negative $\beta^m$ value varies between −11.50 and −1.63, showing high amounts of interactions or synergistic impacts between the mixture components. From the average value of $\beta^m$, it is evident that in the presence of NaCl, the mixture components interact more strongly than in aqueous, U or TU media, hence more negative value of $\beta^m$ was obtained in NaCl media [56]. The effect of NaCl on $\beta^m$ is consistent with the effect on the $cmc$ described earlier. In the presence of NaCl, repulsive interactions among the components of the mixed micelles further decreased. Rubingh's theory predicts that the $\beta^m$ value of any mixed system should be constant despite changes in the component composition, but here the $\beta^m$ value varies with the change in component compositions, showing the limitation of the above theory [6].

Regardless, a negative $\beta^m$ value suggests attractive interactions occur between components, but to verify synergistic effects despite the attractive interactions, one more condition should be obeyed by the system: $|\beta^m| > |\ln(cmc_1/cmc_2)|$. As shown in table 2, all the $\beta^m$ values are below zero but $|\beta^m|$ is higher than $|\ln(cmc_1/cmc_2)|$ for all $\alpha_1$ of BZCl in NaCl media, but only the higher or highest $\alpha_1$ in aqueous, U and TU media. Consequently, synergistic effects between mixture components were attained only at the higher or highest $\alpha_1$ in aqueous, U, TU media, and all $\alpha_1$ in NaCl media. In the remaining cases, only attractive interactions were detected.

Another parameter, the activity coefficients of both components ($f_1^m$ (BZCl) and $f_2^m$ (IMP)), were calculated using the previously obtained values of $X_1^m$ and $\beta^m$ via the following equations:

$$f_1^m = \exp[\beta^m(1 - X_1^m)^2] \tag{3.5}$$

and

$$f_2^m = \exp[\beta^m(X_1^m)^2]. \tag{3.6}$$

Table 2 shows the calculated values of $f_1^m$ and $f_2^m$ in all media. The values of $f_1^m$ as well as $f_2^m$ are less than one in all cases, indicating non-ideal behaviour and interactions between the components [57]. Activity coefficient is a thing used in thermodynamics to describe for deviations from ideal behaviour in a mixed system of chemical ingredients. The activity coefficient is a portion of the excess chemical potential of

ingredients from a mixed system. Table 2 shows that with an increase in the $\alpha_1$ value, the $f_2^m$ value consistently declined, confirming that with increasing in $\alpha_1$, the non-ideal behaviour and interactions between the components increase. This trend explains the difference in the distribution of the components in the mixed micelles.

## 3.3. Properties of IMP + BZCl mixtures at interfacial surfaces

The fundamental properties of pure and mixed amphiphile systems occur at the interfacial surface, affecting the changes in the surface properties of the aqueous system. The number of amphiphilic molecules adsorbed per unit area of the surface is called the maximum surface excess concentration ($\Gamma_{\max}$) and is determined using the Gibbs adsorption equation [58]. The $\Gamma_{\max}$ value in dilute aqueous or non-aqueous solutions is calculated using the Gibbs adsorption isotherm [58].

$$\Gamma_{\max} = -\frac{1}{2.303nRT}\left(\frac{\partial \gamma}{\partial \log(C)}\right)(\text{mol m}^{-2}), \tag{3.7}$$

where $\gamma$, $R$, $T$ and $C$ are the surface tension (mN m$^{-1}$), the ideal gas constant, the temperature ($K$) and the total added concentration of compounds in pure and mixed forms, respectively. $n$ is the entire count number of each species of amphiphile molecule participating in the adsorption process [6], and $n$ is determined for both components (IMP and BZCl). However, in the case of a mixture, the $n$ value is calculated using $n = n_1 X_1^\sigma + n_2(1 - X_1^\sigma)$ [59], where $X_1^\sigma$ is the composition of the first component (i.e. BZCl) at the mixed interface (table 3). For all systems, the value of the slope, $(\partial \gamma/\partial \log(C))$, was determined at a fixed concentration to evaluate $\Gamma_{\max}$.

Once a monolayer is saturated, then an additional layer of the monomer molecules initiates micelle formation. The minimum area occupied by each monomer ($A_{\min}$) on the saturated monolayer is correlated to the $\Gamma_{\max}$ and can be determined from the following equation [58]:

$$A_{\min} = \frac{10^{18}}{N_A\,\Gamma_{\max}}\,(\text{nm}^2), \tag{3.8}$$

where $N_A$ denotes Avogadro's number.

All computed values of $\Gamma_{\max}$ and $A_{\min}$ for all pure compounds and mixtures in all media are presented in table 3. The $\Gamma_{\max}$ value of pure IMP is lower than the $\Gamma_{\max}$ value of pure BZCl, which indicates that $A_{\min}$ is greater for IMP than BZCl as these parameters are inversely proportional, signifying that BZCl is more surface active than IMP. For IMP + BZCl mixtures, the $\Gamma_{\max}$ value in all media except NaCl was either above or close to the $\Gamma_{\max}$ of the pure components. With an increase in the $\alpha_1$ of BZCl in all media except NaCl, the $\Gamma_{\max}$ value decreases with few exceptions (table 3).

Under ideal conditions, the minimum area engaged by each molecule ($A^{\text{id}}$) was calculated using equation (3.9).

$$A^{\text{id}} = X_1^\sigma A_1 + (1 - X_1^\sigma)A_2, \tag{3.9}$$

where $A_1$ is the $A_{\min}$ of pure BZCl and $A_2$ is the $A_{\min}$ of pure IMP. The calculated $A^{\text{id}}$ of IMP + BZCl in aqueous, NaCl, U and TU media is greater than the $A_{\min}$ values at all $\alpha_1$ of BZCl with a few exceptions (table 3). This phenomenon shows that the reduced repulsion between the components with similar head groups is probably responsible for the decreased $A_{\min}$ value. Overall, these results indicate the non-ideal behaviour of the IMP + BZCl mixture in various media.

## 3.4. IMP–BZCl interaction at the interfacial surface

Using Rosen & Hua's theoretical approach [60], the interaction parameters and the compositions of the components at the air–water interface between IMP and BZCl at the formation of the Gibbs monolayer are evaluated. In agreement with the Rubingh model [52], the proportions of the components of the adsorbed mixed monolayer and the interaction parameters at the interface were evaluated by Rosen & Hua's approach as follows [60,61]:

$$\frac{(X_1^\sigma)^2 \ln[(\alpha_1 C/X_1^\sigma C_1)]}{(1 - X_1^\sigma)^2 \ln[(1 - \alpha_1)C/(1 - X_1^\sigma)\,C_2]} = 1 \tag{3.10}$$

**Table 3.** Numerous interfacial parameters for IMP + BZCl mixed systems in different solvents at 298.15 K.[a]

| $\alpha_1$ | $X_1^\sigma$ | $\beta^\sigma$ | $f_1^\sigma$ | $f_2^\sigma$ | $\Gamma_{max}\,10^7$ (mol m$^{-2}$) | $A_{min}/A^{id}$ (nm$^2$) | $\gamma_{cmc}$ | $\pi_{cmc}$ (mN m$^{-1}$) | $pC_{20}$ | $\ln(C_1/C_2)$ |
|---|---|---|---|---|---|---|---|---|---|---|
| aqueous solution | | | | | | | | | | |
| 0 | | | | | 12.8 | 1.30 | 42.6 | 28.4 | 1.95 | |
| 0.1 | 0.560 | −3.65 | 0.493 | 0.319 | 19.2 | 0.87/1.13 | 34.1 | 36.9 | 2.76 | |
| 0.3 | 0.658 | −4.36 | 0.599 | 0.152 | 16.2 | 1.03/1.09 | 35.1 | 35.9 | 3.08 | |
| 0.5 | 0.692 | −5.39 | 0.599 | 0.076 | 15.6 | 1.07/1.08 | 35.6 | 35.4 | 3.28 | −2.87 |
| 0.7 | 0.773 | −4.58 | 0.790 | 0.065 | 16.1 | 1.03/1.06 | 36.4 | 34.7 | 3.26 | |
| 0.9 | 0.747 | −8.14 | 0.594 | 0.011 | 14.0 | 1.18/1.07 | 35.7 | 35.4 | 3.51 | |
| 1 | | | | | 16.8 | 0.99 | 36.8 | 34.2 | 3.20 | |
| 50 mmol kg$^{-1}$ NaCl | | | | | | | | | | |
| 0 | | | | | 8.86 | 1.87 | 44.7 | 26.3 | 2.07 | |
| 0.1 | 0.654 | −7.25 | 0.420 | 0.045 | 10.8 | 1.54/1.83 | 32.9 | 38.1 | 3.83 | |
| 0.3 | 0.808 | −4.52 | 0.847 | 0.052 | 12.1 | 1.37/1.82 | 33.0 | 38.0 | 3.91 | |
| 0.5 | 0.830 | −5.29 | 0.857 | 0.026 | 13.1 | 1.27/1.82 | 32.9 | 38.1 | 4.12 | −5.07 |
| 0.7 | 0.818 | −6.95 | 0.794 | 0.010 | 13.3 | 1.25/1.82 | 32.9 | 38.1 | 4.30 | |
| 0.9 | 0.812 | −9.33 | 0.718 | 0.002 | 13.0 | 1.28/1.82 | 32.8 | 38.2 | 4.46 | |
| 1 | | | | | 9.17 | 1.81 | 32.7 | 38.3 | 4.27 | |
| 300 mmol kg$^{-1}$ urea | | | | | | | | | | |
| 0 | | | | | 16.5 | 1.0 | 45.3 | 25.7 | 1.64 | |
| 0.1 | 0.691 | −3.01 | 0.751 | 0.237 | 18.2 | 0.91/1.20 | 34.2 | 36.8 | 2.73 | |
| 0.3 | 0.743 | −4.61 | 0.738 | 0.078 | 13.7 | 1.21/1.22 | 34.6 | 36.4 | 3.19 | |
| 0.5 | 0.840 | −3.69 | 0.909 | 0.074 | 13.7 | 1.21/1.25 | 35.8 | 35.2 | 3.26 | −4.15 |
| 0.7 | 0.741 | −8.18 | 0.578 | 0.011 | 10.6 | 1.57/1.22 | 35.8 | 35.2 | 3.66 | |
| 0.9 | 0.876 | −5.90 | 0.914 | 0.011 | 13.8 | 1.20/1.26 | 36.0 | 35.0 | 3.50 | |
| 1 | | | | | 12.8 | 1.29 | 36.8 | 34.2 | 3.45 | |

(Continued.)

**Table 3.** (*Continued.*)

| $\alpha_1$ | $X_1^{\sigma}$ | $\beta^{\sigma}$ | $f_1^{\sigma}$ | $f_2^{\sigma}$ | $\Gamma_{\max}\,10^7$ (mol m$^{-2}$) | $A_{\min}/A^{\mathrm{id}}$ (nm$^2$) | $\gamma_{\mathrm{cmc}}$ | $\pi_{\mathrm{cmc}}$ (mN m$^{-1}$) | $pC_{20}$ | $\ln(C_1/C_2)$ |
|---|---|---|---|---|---|---|---|---|---|---|
| **300 mmol kg$^{-1}$ thiourea** | | | | | | | | | | |
| 0 | | | | | 15.4 | 1.08 | 47.4 | 23.6 | 1.54 | |
| 0.1 | 0.693 | −5.04 | 0.622 | 0.089 | 13.6 | 1.22/1.51 | 32.9 | 38.1 | 3.06 | |
| 0.3 | 0.782 | −5.04 | 0.787 | 0.046 | 12.6 | 1.32/1.57 | 32.8 | 38.2 | 3.38 | |
| 0.5 | 0.828 | −5.18 | 0.857 | 0.029 | 11.9 | 1.40/1.60 | 33.5 | 37.5 | 3.54 | −4.96 |
| 0.7 | 0.838 | −6.19 | 0.849 | 0.013 | 12.0 | 1.38/1.60 | 33.3 | 37.7 | 3.69 | |
| 0.9 | 0.817 | −8.94 | 0.741 | 0.003 | 11.5 | 1.45/1.59 | 33.5 | 37.5 | 3.87 | |
| 1 | | | | | 9.74 | 1.71 | 32.6 | 38.4 | 3.99 | |

[a]Standard uncertainties ($u$) are $u(T) = 0.20$ K, $u(\mathrm{NaCl}) = 1$ mmol kg$^{-1}$, $u(\mathrm{urea}) = 2$ mmol kg$^{-1}$, $u(\mathrm{thiourea}) = 2$ mmol kg$^{-1}$ and $u(p) = 5$ kPa (level of confidence = 0.68). Relative standard uncertainties ($u_r$) are $u_r(X_1^{\sigma}) = \pm 2\%$, $u_r(\beta^{\sigma}) = \pm 3\%$, $u_r(f_1^{\sigma}/f_2^{\sigma}) = \pm 4\%$, $u_r(\Gamma_{\max}) = \pm 5\%$, $u_r(A_{\min}/A^{\mathrm{id}}) = \pm 5\%$, $u_r(\pi_{\mathrm{cmc}}) = \pm 2\%$, $u_r(pC_{20}) = \pm 3\%$ and $u_r(\gamma_{\mathrm{cmc}}) = \pm 2\%$.

and

$$\beta^{\sigma} = \frac{\ln\left(C\alpha_1 / C_1 X_1^{\sigma}\right)}{\left(1 - X_1^{\sigma}\right)^2},$$ (3.11)

where $C_1$, $C_2$ and $C$ are the concentrations of pure BZCl, pure IMP and the IMP + BZCl mixture at various $\alpha_1$ in all media of fixed selected surface tension in all cases and $X_1^{\sigma}$ is the mixed monolayer composition of BZCl. The evaluated $X_1^{\sigma}$ and $\beta^{\sigma}$ values of the currently employed system are given in table 3. We measured the $X_1^{\sigma}$ value between 0.5598 and 0.8761; the interfacial surface contains a large amount of BZCl. Tables 2 and 3 show that the average $X_1^{\sigma}$ value feel in a range similar to the average $X_1^{m}$ value, which suggests that both the mixed monolayer and the mixed micelles contain a high concentration of BZCl. These $X_1^{\sigma}$ values do not signify any regular increase in the $\alpha_1$ of BZCl, but overall they are higher at higher $\alpha_1$. Compared with the aqueous medium, the $X_1^{\sigma}$ value is greater for the NaCl medium, as NaCl screens the repulsive interactions between IMP and BZCl so that more BZCl monomers are incorporated in the mixed monolayer (table 3).

Akin to $\beta^{m}$, negative, positive and zero (or close to zero) values of $\beta^{\sigma}$ indicate attractive interaction, repulsion interaction, and neither interaction nor repulsion (ideal mixing), respectively, among the components of a mixed monolayer [6]. In this work, all the $\beta^{\sigma}$ values are negative, indicating attractive interactions between the component monomers at the interface (table 3). In all media, the average value of $\beta^{\sigma}$ in all systems is greater than the average value of $\beta^{m}$, signifying the interactions between components are greater in the mixed monolayer than the mixed micelles. The $\beta^{\sigma}$ values in the NaCl are the most negative among all the media, suggesting that, in the presence of NaCl, the interaction between the constituents increased at the monolayer.

The mixtures of IMP and BZCl display higher surface properties and considerably lower *cmc* values than pure IMP. The synergism in the binary mixed systems also depends on the associated characteristics of each component other than the interaction strength ($\beta^{m}$ or $\beta^{\sigma}$). The states for synergism at the interfacial surface which reduce surface tension efficiency are (a) the total required concentration of the mixture of the components to reduce the surface tension of water or any other media to the 20 mN m$^{-1}$, as the surface tension should be lower than that of a particular component at the interface, and (b) the mixed system subsequently obeys the following conditions [6]:

(i) $\beta^{\sigma} <$ zero
(ii) $|\beta^{\sigma}| > |\ln\left(C_1/C_2\right)|$

Table 3 shows that the $\beta^{\sigma}$ values were negative in all cases and the value of $|\beta^{\sigma}|$ was greater than the value of $|\ln\left(C_1/C_2\right)|$ for all systems with a few exceptions. Therefore, the IMP + BZCl mixed system showed synergism in surface tension reduction efficiency.

As for the mixed micelles, the activity coefficients of both constituents of the formed monolayer, $f_1^{\sigma}$ (BZCl) and $f_2^{\sigma}$ (IMP), were also calculated using $\beta^{\sigma}$ and $X_1^{\sigma}$ through the following equations [6].

$$f_1^{\sigma} = \exp\{\beta^{\sigma}(1 - X_1^{\sigma})^2\},$$ (3.12)

and

$$f_2^{\sigma} = \exp\{\beta^{\sigma}(X_1^{\sigma})^2\}.$$ (3.13)

Table 3 shows the obtained $f_1^{\sigma}$ and $f_2^{\sigma}$ values of the mixed systems in all media, and in every case, are below 1, indicating attractive interactions among the components at the interface as well as non-ideal behaviour. Table 3 also shows the involvement of BZCl is greater than the involvement of IMP in all media at the monolayer, as $f_1^{\sigma} > f_2^{\sigma}$ in each system and the involvement of IMP at the monolayer decreased with increases in $\alpha_1$.

Another parameter, which is frequently applied to assess the efficiency of the adsorption of amphiphile solutions, is $pC_{20}$, which is determined using equation (3.14) [6]:

$$pC_{20} = -\log C_{20},$$ (3.14)

where $pC_{20}$ is the concentration required to lessen the surface tension of aqueous or non-aqueous systems by 20 mN m$^{-1}$. The higher the $pC_{20}$ value, the more effectively the amphiphiles are adsorbed at the interfacial surface, and the more effectively it lessens surface tension. This effect determines the concentrations of amphiphiles essential to achieve saturation adsorption or decrease the surface tension by 20 mN m$^{-1}$. The calculated $pC_{20}$ values of pure IMP, BZCl and IMP + BZCl mixtures are

given in table 3. The results reveal that the computed value of $pC_{20}$ of BZCl is greater than that for IMP in each medium, showing that the surfactant has greater surface adsorption efficiency than IMP. In addition, their corresponding *cmc* values, determined previously, indicate this. In the case of a mixed system (IMP + BZCl), the surface adsorption efficiency was greater than pure IMP, and with an increase in $\alpha_1$, a considerable increase in $pC_{20}$ value is seen, which means the surface adsorption efficiency of mixed systems increases with an increase in $\alpha_1$. However, $pC_{20}$ for the IMP + BZCl mixture was less than the $pC_{20}$ of BZCl at lower $\alpha_1$ and close to the $pC_{20}$ of BZCl at 0.5 $\alpha_1$. At higher $\alpha_1$, the $pC_{20}$ value of the mixture surpasses the $pC_{20}$ value of both components (table 3).

The capability of an amphiphile to lessen surface tension is assessed via the extent of reduction, or the surface pressure ($\pi_{cmc} = (\gamma_0 - \gamma_{cmc})$), at the *cmc*, because the reduction of the surface tension beyond the *cmc* is somewhat unimportant. Here, $\gamma_0$ describes the surface tension of pure aqueous or non-aqueous media, and $\gamma_{cmc}$ indicates the $\gamma$ at the *cmc* of a system (pure or mixed). Table 3 presents the $\gamma_{cmc}$ and $\pi_{cmc}$ of all the systems studied. IMP has the highest $\gamma_{cmc}$, while pure BZCl and IMP + BZCl mixtures have similar values (table 3). The $\pi_{cmc}$ value of IMP was less than that of pure BZCl in each medium; but the $\pi_{cmc}$ of the IMP + BZCl mixtures is greater than the $\pi_{cmc}$ of IMP and less than the $\pi_{cmc}$ of BZCl.

## 3.5. Thermodynamic parameters

It is important to evaluate the micellar solution thermodynamic parameters in the aqueous and non-aqueous systems because they indicate the comparative significance of hydrophobic interactions, water associated with amphiphiles, as well as head group repulsions. The Gibbs free energy ($\Delta G_{\mathrm{mic}}^{\circ}$) of micellization is evaluated using equation (3.15) [62].

$$\Delta G_{\mathrm{mic}}^{0} = RT \ln X_{cmc}. \tag{3.15}$$

Here, the degree of ionization is considered one [6] because tensiometry cannot measure the degree of ionization; therefore the ionization of amphiphiles is considered complete. $X_{cmc}$ is the *cmc* in mole fraction, and $T$ and $R$ have their usual meanings.

The computed $\Delta G_{\mathrm{mic}}^{\circ}$ values for pure IMP, BZCl and IMP + BZCl mixtures in various ratios in each medium are given in table 4 and show that their values are negative in every case. The negative values suggest that the association process is spontaneous and thermodynamically stabilized. The larger the magnitude of $\Delta G_{\mathrm{mic}}^{\circ}$, showed the greater the spontaneity of the aggregation process. Through an increase in the $\alpha_1$ of BZCl in solution, the magnitude of $\Delta G_{\mathrm{mic}}^{\circ}$ increases regularly and reaches the maximum negative value at the maximum $\alpha_1$ tested, indicating that the mixed systems become more spontaneous and thermodynamically stabilized with an increase in $\alpha_1$ [63]. The $\Delta G_{\mathrm{mic}}^{\circ}$ value of pure IMP is in good agreement with earlier reported work [64]. The $\Delta G_{\mathrm{mic}}^{\circ}$ value for pure BZCl is also in good agreement with previously measured values [40,65]. Table 4 also shows that for the BZCl surfactant, the $\Delta G_{\mathrm{mic}}^{\circ}$ value was greater than the $\Delta G_{\mathrm{mic}}^{\circ}$ value of IMP, which means the association of BZCl is more spontaneous because BZCl has more hydrophobicity than IMP. Table 4 also shows that in the NaCl media, the values of the $\Delta G_{\mathrm{mic}}^{\circ}$ of the mixed systems are more negative than in the aqueous system, indicating that in the presence of NaCl, the hydrophobicity of pure as well as mixed systems is enhanced as the interactions between similar and dissimilar molecules are raised, and the electrostatic repulsions are reduced. As a result, the pure or mixed association process starts at a lower concentration than in the aqueous system. Instead, the $\Delta G_{\mathrm{mic}}^{\circ}$ value of the tested mixture is less negative in U or TU media than in aqueous solution, revealing that the interactions among similar and dissimilar molecules are weaker in the U or TU media. The aggregation behaviour of compounds remains spontaneous in U or TU media; however, the spontaneity decreases to some extent compared with the other systems studied (table 4). As the concentration of U and TU used is the same, but in TU media, the system is less spontaneous than in U media (i.e. the magnitude of $\Delta G_{\mathrm{mic}}^{\circ}$ is lowest in TU media). It is concluded from the overall results that the magnitude of $\Delta G_{\mathrm{mic}}^{\circ}$ is inversely proportional to the *cmc* value of the corresponding system. In TU medium, the *cmc* of pure compounds and mixtures is greater than in aqueous, NaCl and U media, and the magnitude of $\Delta G_{\mathrm{mic}}^{\circ}$ is greater in the U medium than in the other media used. The negative $\Delta G_{\mathrm{mic}}^{\circ}$ value of the studied system in different media are found in the following order:

$$\Delta G_{\mathrm{mic}}^{0}(\mathrm{NaCl}) > \Delta G_{\mathrm{mic}}^{0}(\mathrm{aqueous}) > \Delta G_{\mathrm{mic}}^{0}(\mathrm{U}) > \Delta G_{\mathrm{mic}}^{0}(\mathrm{TU}).$$

**Table 4.** Numerous thermodynamic parameters and packing parameter (P) for IMP + BZCl mixed systems in various solvents at 298.15 K.[a]

| $\alpha_1$ | $\Delta G^{\circ}_{mic}$ (kJ mol$^{-1}$) | $\Delta G^{\circ}_{ad}$ (kJ mol$^{-1}$) | $G_{min}$ (kJ mol$^{-1}$) | $\Delta G^{m}_{ex}$ (kJ mol$^{-1}$) | $\Delta G^{cr}_{ex}$ (kJ mol$^{-1}$) | $p$-value |
|---|---|---|---|---|---|---|
| aqueous system | | | | | | |
| 0 | −17.8 | −40.1 | 33.3 | | | 0.34 |
| 0.1 | −21.3 | −40.5 | 17.8 | −1.43 | −2.23 | 0.50 |
| 0.3 | −22.6 | −44.8 | 21.7 | −0.98 | −2.44 | 0.42 |
| 0.5 | −23.7 | −46.4 | 22.9 | −1.15 | −2.85 | 0.41 |
| 0.7 | −23.9 | −45.4 | 22.6 | −0.52 | −1.99 | 0.42 |
| 0.9 | −24.6 | −49.8 | 25.4 | −1.01 | −3.81 | 0.37 |
| 1 | −24.4 | −44.7 | 21.9 | | | 0.45 |
| 50 mmol kg$^{-1}$ NaCl | | | | | | |
| 0 | −18.1 | −47.8 | 50.4 | | | 0.24 |
| 0.1 | −23.3 | −58.6 | 30.4 | −3.10 | −4.06 | 0.28 |
| 0.3 | −24.9 | −56.3 | 27.2 | −3.21 | −1.74 | 0.32 |
| 0.5 | −26.7 | −55.9 | 25.1 | −4.46 | −1.86 | 0.34 |
| 0.7 | −27.7 | −56.3 | 24.6 | −5.18 | −2.57 | 0.35 |
| 0.9 | −28.4 | −57.8 | 25.3 | −6.23 | −3.54 | 0.34 |
| 1 | −24.8 | −66.6 | 35.7 | | | 0.24 |
| 300 mmol kg$^{-1}$ urea | | | | | | |
| 0 | −17.6 | −33.2 | 27.4 | | | 0.44 |
| 0.1 | −20.9 | −41.1 | 18.8 | −1.29 | −1.59 | 0.48 |
| 0.3 | −22.2 | −48.7 | 25.2 | −0.83 | −2.18 | 0.36 |
| 0.5 | −23.0 | −48.6 | 26.1 | −0.67 | −1.23 | 0.36 |
| 0.7 | −23.6 | −56.9 | 33.8 | −0.62 | −3.89 | 0.28 |
| 0.9 | −24.4 | −49.8 | 26.2 | −1.34 | −1.59 | 0.36 |
| 1 | −24.1 | −50.7 | 28.7 | | | 0.34 |

(Continued.)

**Table 4.** (Continued.)

| $\alpha_1$ | $\Delta G_{mic}^{\circ}$ (kJ mol$^{-1}$) | $\Delta G_{ad}^{\circ}$ (kJ mol$^{-1}$) | $G_{min}$ (kJ mol$^{-1}$) | $\Delta G_{ex}^{m}$ (kJ mol$^{-1}$) | $\Delta G_{ex}^{\sigma}$ (kJ mol$^{-1}$) | $p$-value |
|---|---|---|---|---|---|---|
| 300 mmol kg$^{-1}$ thiourea | | | | | | |
| 0 | −17.58 | −32.9 | 30.8 | | | 0.41 |
| 0.1 | −20.81 | −48.8 | 24.2 | −1.26 | −2.65 | 0.36 |
| 0.3 | −22.07 | −52.3 | 26.0 | −0.79 | −2.13 | 0.33 |
| 0.5 | −22.91 | −54.5 | 28.3 | −0.65 | −1.83 | 0.31 |
| 0.7 | −23.54 | −54.8 | 27.7 | −0.61 | −2.09 | 0.32 |
| 0.9 | −24.34 | −57.1 | 29.2 | −1.33 | −3.31 | 0.30 |
| 1 | −23.98 | −63.4 | 33.5 | | | 0.26 |

[a]Standard uncertainties ($u$) are $u(T) = 0.20$ K, $u(NaCl) = 1$ mmol kg$^{-1}$, $u(urea) = 2$ mmol kg$^{-1}$, $u(thiourea) = 2$ mmol kg$^{-1}$ and $u(p) = 5$ kPa (level of confidence = 0.68). Relative standard uncertainties ($u_r$) are $u_r(\Delta G_m^{\circ}) = \pm 3\%$, $u_r(\Delta G_{ads}^{\circ}) = \pm 4\%$, $u_r(G_{min}) = \pm 4\%$, $u_r(\Delta G_{ex}^{m}/\Delta G_{ex}^{\sigma}) = \pm 5\%$ and $u_r(p) = \pm 4\%$.

One additional thermodynamic parameter is the standard free energy of adsorption ($\Delta G_{ad}^{\circ}$) of the pure and mixed amphiphiles system at the air–water interface as calculated from equation (3.16) [66,67].

$$\Delta G_{ad}^{\circ} = \Delta G_{mic}^{\circ} - \frac{\pi_{cmc}}{\Gamma_{max}}. \tag{3.16}$$

Table 4 shows that the $\Delta G_{ad}^{\circ}$ values for the pure and mixed systems in each medium were negative, illustrating that the adsorption of constituents at the interface occurs spontaneously. Additionally, the value of $\Delta G_{ad}^{\circ}$ is much more negative than the value of $\Delta G_{mic}^{\circ}$, signifying that the association is a secondary process with respect to interfacial adsorption; therefore, work must be required to convert monomers from the monomeric form to the micellar form. This difference also indicates that the adsorption process is more favourable [68]. Additionally, the higher magnitude of $\Delta G_{ad}^{\circ}$ shows that the adsorption process is preferred over the micellization process in bulk systems because the hydrophobic portion of molecules always prefers the interface (table 4). In all media, $\Delta G_{ad}^{\circ}$ of pure BZCl is greater than $\Delta G_{ad}^{\circ}$ for pure IMP, showing that adsorption in BZCl is easier and comparatively more spontaneous than IMP. However, for IMP + BZCl mixtures, the $\Delta G_{ad}^{\circ}$ value was greater than the $\Delta G_{ad}^{\circ}$ value of IMP but less than the $\Delta G_{ad}^{\circ}$ value of pure BZCl. Overall, the $\Delta G_{ad}^{\circ}$ value shows that the adsorption process in the case of IMP + BZCl mixtures is easier than pure IMP adsorption. In NaCl media, the value of $\Delta G_{ad}^{\circ}$ of all systems is more negative than in the aqueous system. This rise in the magnitude of $\Delta G_{ad}^{\circ}$ in NaCl media is due to the diminished electrostatic repulsions between the molecules. Consequently, more molecules can reside at the surface, which enriches the density of the monomers at the monolayer, and hence additional work is needed to transport more monomers from the surface to the micellar form than in the aqueous system.

Sugihara et al. [69] have computed the minimum free energy of the interface at the maximum adsorption ($G_{min}$) achieved at cmc, another thermodynamic parameter, using equation (3.17).

$$G_{min} = \gamma_{cmc} A_{cmc} N_A. \tag{3.17}$$

The values calculated for all of the systems studied are given in table 4. The smaller value of $G_{min}$ displays the more stable interfacial surface formation; this parameter is directly related to the attractive interactions among monomers. In this work, smaller values of $G_{min}$ are achieved in the case of pure compounds and their mixtures in each medium, showing that a thermodynamically stable surface is fashioned. This requires that interactions among components are valued. The $G_{min}$ of the mixed system at each $\alpha_1$ of their respective media is less than the pure compound in the same medium, verifying that a mixed monolayer is more thermodynamically stable than a single component monolayer (table 4).

An additional important thermodynamic parameter called excess free energy ($\Delta G_{ex}$) is computed by applying equations (3.18) and (3.19) [70–73].

$$\Delta G_{ex}^{m} = RT[X_1^{m} \ln f_1^{m} + (1 - X_1^{m}) \ln f_2^{m}] \tag{3.18}$$

and

$$\Delta G_{ex}^{\sigma} = RT[X_1^{\sigma} \ln f_1^{\sigma} + (1 - X_1^{\sigma}) \ln f_2^{\sigma}]. \tag{3.19}$$

In equations (3.18) and (3.19), $\Delta G_{ex}^{m}$ and $\Delta G_{ex}^{\sigma}$ denote the excess free energy of mixed micelles and mixed monolayers, respectively, and their values in each medium are given in table 4. All the $\Delta G_{ex}$ are negative, demonstrating that the stability of the mixed micelles, as well as the mixed monolayers, is greater for mixtures than pure systems because of the interactions between the different components. Mixed micelles and mixed interfaces have distinct physico-chemical characteristics from micelles/interfaces formed by single species. From the perspective of applications, mixed micelles and mixed interfaces show immense significance in different fields (such as pharmaceutical, enhanced oil recovery activity, biological, solubilization of hydrophobic molecules and dispersion). The cmc values of binary mixtures were much lower than those of either component alone, and the interfacial properties of the binary mixed system were greatly enhanced. These qualities are of significant value, as the use of mixed components systems for any purpose leads to lower costs as well as lower environmental impact. Consequently, mixed micelles and mixed monolayers formed by binary components mixtures are more stable than micelles and monolayers formed from either of the pure components. Overall, $\Delta G_{ex}^{m}$ and $\Delta G_{ex}^{\sigma}$ do not follow any specific trend with increased $\alpha_1$, even in different media. Through comparing the $\Delta G_{ex}^{\sigma}$ and $\Delta G_{ex}^{m}$ values, the average $\Delta G_{ex}^{\sigma}$ value is greater than the average $\Delta G_{ex}^{m}$ value, implying that the mixed monolayer shows additional stability (table 4). Herein, it is found that the value of

interaction parameters was not constant through the change in $\alpha_1$ of ingredients which means that the involved $\Delta G_{ex}^m$ will not be symmetrical, or it can say that molecular interactions diverge from symmetrical solutions displaying the limitations of Rubingh's model [52,74].

## 3.6. Packing parameters of IMP + BZCl

The micellar shape generated in an aqueous or non-aqueous system is significant in evaluating various properties of the amphiphile solutions, such as viscosity, capability to solubilize water-insoluble hydrophobic compounds, and phase separation [6]. The packing parameter ($P$) is used to evaluate the shape of the micelles formed in an aqueous or non-aqueous system using Israelachvili et al.'s equation [75].

$$P = \frac{V_o}{A_{min}l_c},$$

(3.20)

where $V_0$ is the volume engaged through the hydrophobic groups in the core of the micelles, $l_c$ is the hydrophobic chain length in the micellar core and $A_{min}$ is the minimum area of the hydrophilic group at the interfacial surface. $V_o = [27.4 + 26.9\,(n_c - 1)] \times 2$ (Å$^3$); and $l_c = [1.54 + 1.26\,(n_c - 1)]$ (Å). $V_o$ and $l_c$ are evaluated using the formula given by Tanford [76], where $n_c$ is the total number of C atoms in the hydrocarbon chain. $n_c$ is considered less than the actual count of C atoms as the C atom which was connected directly with the head group, which is highly solvated, so this is also considered as part of the head group [77].

The calculated $p$-values of the pure (IMP and BZCl) and mixed systems (IMP + BZCl) in all the media studied are presented in table 4. Moreover, from the literature, a $p$-value between 0 and 1/3 will be obtained for spherical micelles, between 1/3 and 1/2 signifies cylindrical micelles, and between 0.5 and 1 shows vesicular micelles [6]. Results in table 4 indicate that the $p$-value for pure IMP in NaCl medium is 0.24, indicating spherical micelles, while in aqueous, U and TU media, the $p$-value is between 0.33 and 0.5, indicating the formation of cylindrical micelles. In the case of BZCl in NaCl and TU, the $p$-values are less than 0.24 and 0.26, respectively, due to the formation of spherical micelles, while in aqueous and U media, the $p$-values are 0.45 and 0.34, respectively, indicating cylindrical micelles have formed. In the IMP + BZCl mixture, the $p$-value is between 0.33 and 0.50 in most cases and is attributable to vesicular micelles. However, in some other cases, the value is less than 0.33, indicating the formation of spherical micelles.

## 3.7. FTIR spectroscopy

FTIR spectroscopy is used to explore the interactions between components of the micellar solutions [78,79]. In particular, the frequencies of the hydrophobic and head group parts of the amphiphiles deliver information about structural changes in the micellar monomers [80]. Background-subtracted FTIR spectra of pure IMP in aqueous solution and the 1:1 IMP + BZCl mixed system are shown in figure 3a,b. The interaction of IMP with BZCl is possibly observed in the shifting of C–N stretching, C–H bending and stretching of the head of IMP. Figure 3a shows FTIR spectra of IMP and IMP + BZCl mixtures from 1090 to 1490 cm$^{-1}$ to assess the influence of BZCl on C–N stretching (aliphatic) along with C–H bending in IMP monomers. IMP is cationic, contains three alkyl groups, and shows C–N stretching (aliphatic) at three different frequencies. These vibrational bands occur at 1109.18, 1212.18 and 1224.91 cm$^{-1}$ in the spectrum of the pure compound, but in the presence of BZCl, that is, for the IMP + BZCl mixtures, these obtained frequencies (C–N stretching) shift to higher frequencies of 1110.49, 1212.55 and 1225.88 cm$^{-1}$. In IMP, C–H bending is detected at three distinct frequencies: 1398.85, 1446.61 and 1486.60 cm$^{-1}$; in IMP + BZCl mixtures, the frequencies of C–H bending in IMP shift to 1395.45, 1447.33 and 1487.21 cm$^{-1}$.

For further analysis of the interaction of IMP with BZCl, figure 3b shows the frequency range from 2865 to 2980 cm$^{-1}$ to view the effect of BZCl on the frequency of the C–H stretching of IMP. IMP showed C–H stretching at three frequencies: 2886.74, 2931.84 and 2954.15 cm$^{-1}$. But in the case of the IMP + BZCl mixed system, the C–H stretching bands in IMP move to 2887.61, 2932.72 and 2953.52 cm$^{-1}$. Therefore, in the presence of BZCl, the observed shifts in C–N stretching, C–H stretching, as well as the bending frequency in the head group of IMP indicate the interaction between IMP and BZCl [81].

The FTIR spectra of pure BZCl and 1:1 BZCl + IMP were recorded between 1050 and 1285 cm$^{-1}$, and their spectra are given in figure 3c. Pure BZCl shows the C–O stretching (aliphatic ether and alkyl aryl ether) bands at different frequencies. The C–O stretching (aliphatic ether) bands in BZCl are detected at 1063.67, 1115.02 and 1123.21 cm$^{-1}$, and in the presence of IMP (BZCl + IMP), the position of the

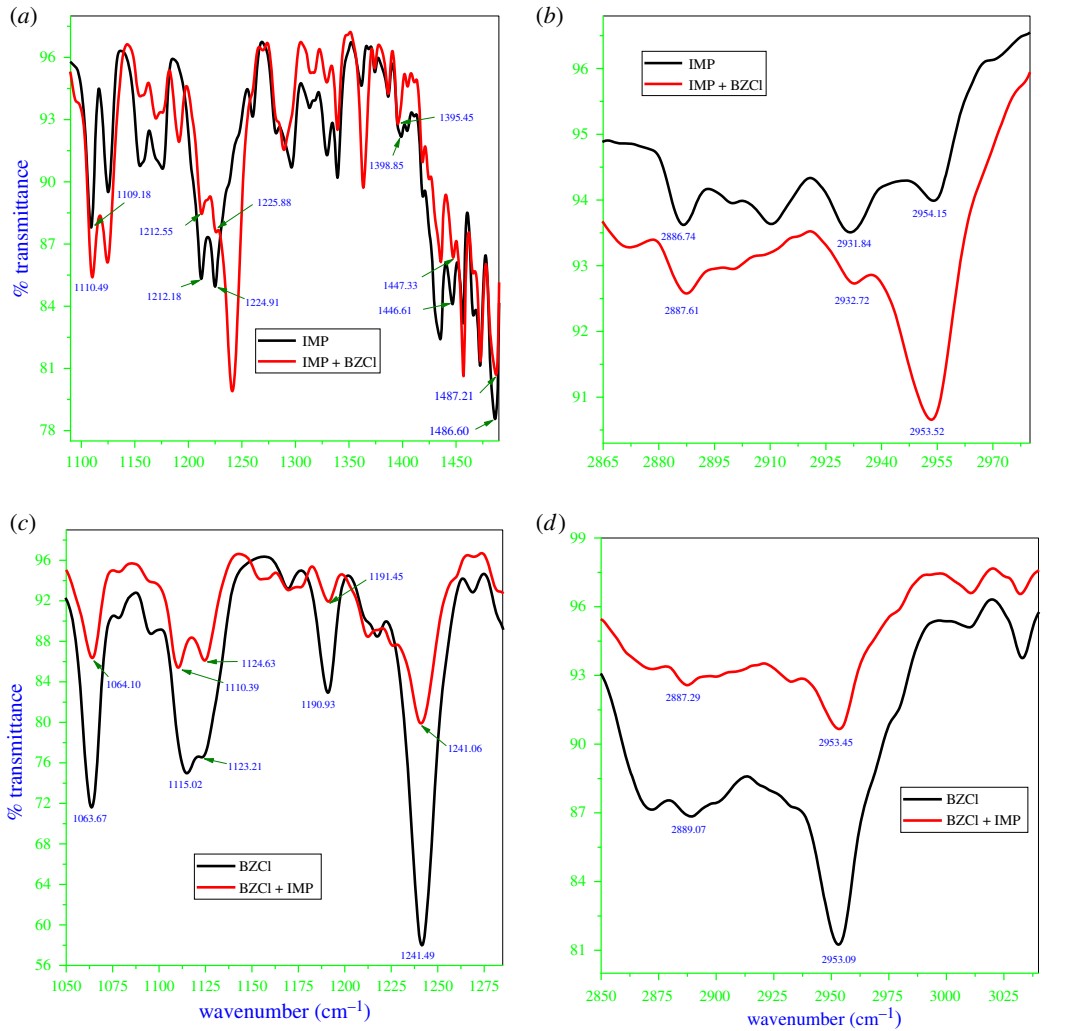

**Figure 3.** FTIR spectra of pure IMP and in existence of BZCl ((*a*) and (*b*)), and BZCl spectra in pure form and presence of IMP ((*c*) and (*d*)).

C–O stretching (aliphatic ether) bands in BZCl is shifted to different wavelengths: 1064.10, 1110.39 and 1124.63 cm$^{-1}$ respectively. The C–O stretching (alkyl aryl ether) bands in BZCl appear at 1190.93 and 1241.49 cm$^{-1}$. However, in the case of the BZCl + IMP mixed system, the C–O stretching (alkyl aryl ether) bands shift from their original positions to 1191.45 and 1241.06 cm$^{-1}$ (figure 3*c*). The shifting in the frequency of the C–O stretching (aliphatic ether and alkyl aryl ether) bands in BZCl in the presence of IMP shows the clear-cut interaction between the components [82].

The frequency range of 2865 to 2980 cm$^{-1}$ also shows the effect of IMP on the C–H stretching bands (methyl and/or methylene) in BZCl. Their spectra are shown in figure 3*d*. The C–H stretching bands in BZCl are present at three different wavelengths (2886.74, 2931.84 and 2954.15 cm$^{-1}$). However, in the BZCl + IMP mixed system, the C–H stretching band of BZCl is changed to different frequencies: the first two frequencies increase, and the third decreases (that is, 2886.74 to 2887.61 cm$^{-1}$, 2931.84 to 2932.72 cm$^{-1}$, 2954.15 to 2953.52 cm$^{-1}$). These frequency shifts show the interactions between the components of the mixed micelles (BZCl + IMP). Because of these interactions, the shifts in frequency in the mixed system compared with the pure system were small but reproducible [83]. Overall, the shifting in the C–N stretching (aliphatic), C–H bending and stretching, and C–O stretching (aliphatic ether and alkyl aryl ether) frequencies in the mixed system compared with the pure systems signifies the interaction between the components.

## 3.8. UV–visible spectroscopy

UV–visible spectroscopy is also employed to investigate the interactions between the constituent compounds of the mixture. A fixed concentration (0.11 mmol kg$^{-1}$) of the amphiphilic drug IMP is

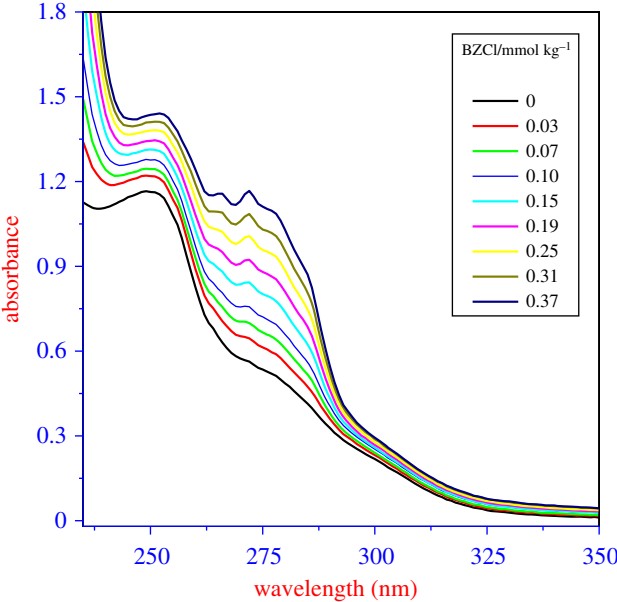

**Figure 4.** UV–visible spectra of pure IMP and in occurrence of different concentration of BZCl in different media ((a) water, (b) 50 mmol kg$^{-1}$ NaCl, (c) 300 mmol kg$^{-1}$ urea and (d) 300 mmol kg$^{-1}$ thiourea).

used for UV–Vis measurements. The titration of IMP (1.5 ml in quartz cuvette) was performed by increasing the concentration of BZCl. To prevent dilution, the stock solution of surfactant (BZCl) (5 mol kg$^{-1}$) was prepared in the 0.11 mmol kg$^{-1}$ IMP solution. First, the absorbance spectrum of the 0.11 mmol kg$^{-1}$ IMP solution was recorded, which shows an absorbance maximum at 249 nm (figure 4). This obtained maximum absorbance wavelength is a π–π* transition. Figure 4 also shows that when BZCl is added to the IMP solution, the absorbance intensity increases. With further increases in the BZCl concentration (from 0.03 to 0.37 mmol kg$^{-1}$) in the IMP solution, the absorbance intensity again increases, confirming that the hyperchromic effect arises because of the attractive interaction between IMP and BZCl [84,85]. However, it shows that at a lower added concentration of BZCl, the maximum absorbance peak of IMP does not change, but at a higher concentration, the peak shifts to a somewhat higher concentration. This indicates that the redshift of a mixed system occurs due to greater interaction between mixture components at higher concentrations (figure 4). At the higher concentration of BZCl in the solution, mixtures also showed maxima at around 272 nm. Overall, the titration results show the interaction between IMP and BZCl, since the absorbance peak of IMP disappears to some extent because of the formation of the complex [86]. An insignificant redshift (2–3 nm) in the maxima of IMP in the presence of BZCl does not describe the complex formation between IMP and BZCl. Therefore, for quantitative evaluation of the binding of BZCl with IMP monomers, absorbance statistics are applied using the Benesi–Hildebrand equation [87,88]

$$\frac{1}{(A - A_0)} = \frac{1}{K(A_{max} - A_0)[BZCl]} + \frac{1}{(A_{max} - A_0)}, \tag{3.21}$$

where $K$ is the binding constant and $A_0$, $A$ and $A_{max}$ are the absorbance values of IMP in the absence, presence and infinite concentration of BZCl, respectively.

The plot of $1/(A - A_0)$ versus $1/[BZCl]$ shows straight lines (figure 5), which is more evidence in favour of the formation of a 1 : 1 IMP : BZCl complex. The $K$ value for the IMP + BZCl complexes is determined as 2510 mol kg$^{-3}$ by evaluating the ratio of the intercept and the slope of the Benesi–Hildebrand plot. The correlation coefficient ($r$) is 0.998, which supports the good linear fit. The obtained value of $K$ is further used to determine the free energy change ($\Delta G$) using equation (3.22) [89].

$$\Delta G = -RT \ln K. \tag{3.22}$$

The $\Delta G$ value obtained for the IMP + BZCl mixed system is negative (−19.4 kJ mol$^{-1}$), which confirms the spontaneity of the complexation process. This may be caused by a reduction in self-electrostatic repulsion among the cationic head group molecules, which shows preferential binding between

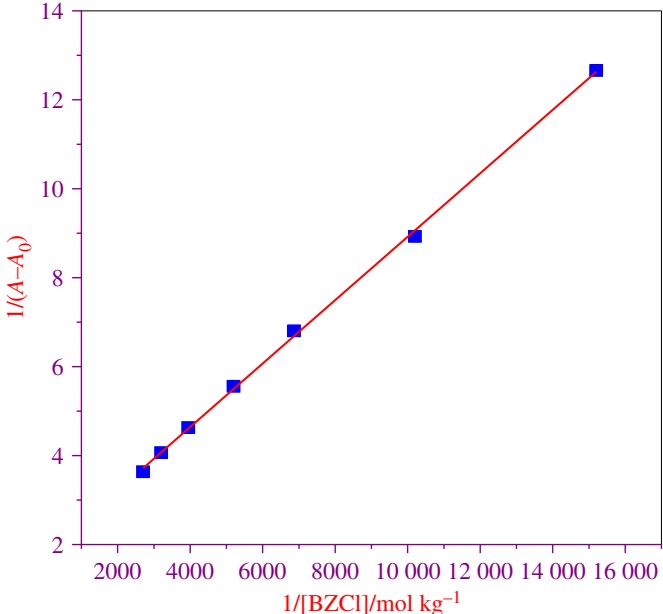

**Figure 5.** The plot of $1/(A-A_0)$ against $1/[BZCl]$ for the interaction of IMP with BZCl.

components. At 0.1 $\alpha_1$ of BZCl, the value of $\Delta G_{mic}^\circ$ was found to be $-21.3\,\text{kJ}\,\text{mol}^{-1}$ showing the micellization process was more spontaneous as compared with the complexation process.

## 4. Conclusion

Before, establishing a surfactant as a suitable drug vehicle, a broad investigation must be completed to inspect and understand the aggregation behaviours of the surfactant with the projected drug. Herein, the mixed micellization behaviour of the IMP drug and BZCl surfactant mixed system in aqueous and other media is investigated via the tensiometry technique. The relationship between $cmc$ and $cmc^{id}$ shows non-ideal behaviour, indicating the presence of attractive interactions and synergism between the components. In NaCl media, the $cmc$ of pure components and mixtures decreases compared with the aqueous system, and in U or TU media, the $cmc$ values increase compared with the aqueous solution. Between U and TU, TU is more effective in increasing the $cmc$ of the system. The $X_1^m$ values were found larger than the employed $\alpha_1$ of BZCl except $\alpha_1 = 0.9$, revealing the higher BZCl contribution in the mixed micelles. The negative values of $\beta^m$ and $\beta^\sigma$ showed the attractive interactions among components in mixed micelles and the mixed monolayers, respectively. The activity coefficients of both components in solution and at the surface were found to be less than 1, showing the non-ideal behaviour and interaction between components. The $\Gamma_{max}$ value for IMP is found lower than BZCl, implying that BZCl has an additional surface active than IMP. In all media, the $pC_{20}$ value for BZCl is higher than IMP, revealing that the BZCl has more surface adsorption efficiency as compared with IMP. The $\Delta G_{mic}^\circ$ value is negative in all media, which is indicative of the spontaneity in the pure and mixed systems, and their spontaneity rises with increased $\alpha_1$; however, $\Delta G_{ad}^\circ$ is greater than the corresponding $\Delta G_{mic}^\circ$ in all media. $\Delta G_{mic}^\circ$ and $\Delta G_{ad}^\circ$ are larger in NaCl media than the other media tested in this study. Excess free energies ($\Delta G_{ex}^m$ and $\Delta G_{ex}^\sigma$) showed that the mixed interfacial surface of mixed micelles is more stable than singular component micelles and monolayers. The FTIR spectra of aqueous IMP in the presence of BZCl or vice versa showed frequencies shifting from their original position owing to the interaction among constituent compounds. UV–Vis spectra display the interaction between IMP and BZCl as shifts in the absorbance maxima of IMP. The results of this study show a straightforward approach to the design of BZCl, which possibly will be an effective drug delivery agent for antidepressant drugs.

Data accessibility. Data that support this study have been uploaded as electronic supplementary material [90].

Authors' contributions. D.K. and M.A.R. did the experiments and wrote the manuscript. Y.G.A. and A.M.A. analysed and interpreted data. All authors gave final approval for publication and agreed to be held accountable for the work performed therein.

Competing interests. The authors declare no competing interest.

Funding. This project was funded by the Deanship of Scientific Research (DSR) at King Abdulaziz University, Jeddah, under grant no. (G:102-130-1441). The authors, therefore, acknowledge with thanks DSR for technical and financial support.

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
