## [Peer Review File · Royal Society Open Science]

Review History

RSOS-211204.R0 (Original submission)

Review form: Reviewer 1

Is the manuscript scientifically sound in its present form?

No

Are the interpretations and conclusions justified by the results?

No

Is the language acceptable?

No

Do you have any ethical concerns with this paper?

No

Have you any concerns about statistical analyses in this paper?

No

Recommendation?

Reject

Comments to the Author(s)

This i'm afraid a poor paper, which is a pity as it is a potentially interesting subject. It is poorly written and contains a lot of background and explanation which to an informed reader should not be necessary. It contains the type of information that a thesis may contain and reads like a poor precis from someone's thesis. In general it come across as naive with an unawareness of the current state of the art in this area..

The data is limited and poorly presented, why, for exaxmple, are the variations in cmc for the mixtures not plotted. There seems to be no error bars in the data or the derived parameters, and indeed to quote many of those parameters to 2 decimal places is simply unrealistic. The thermodynamic analysis using RST but with an interaction parameters which varies with composition is simply not valid. The authors seem to be woefully unaware of recent applications of the pseudo phase approximation in which the asymmetry in the mixing, outside RST, is properly accounted for by expanding the excess free energy of mixing to higher order terms. Also the authiors seem unaware of the limitations of the primarily surface tension approach and that the state of the art has now moved on considerably with the inclusions of new experimental approahces.

Review form: Reviewer 2**Is the manuscript scientifically sound in its present form?**

Yes

Are the interpretations and conclusions justified by the results?

Yes

Is the language acceptable?

No

Do you have any ethical concerns with this paper?

No

Have you any concerns about statistical analyses in this paper?

No

Recommendation?

Accept with minor revision (please list in comments)

Comments to the Author(s)

Authors focus on the interaction behavior between imipramine hydrochloride and benzethonium chloride in different media using different techniques. This work is interesting. The paper can be recommended to be published in this Journal after minor revisions are made as follows.

1. The English language in the whole manuscript should be polished.
2. The abstract should be improved.
3. In introduction, authors should give some explanations to select these materials (NaCl, urea, and thiourea) in this investigation. Also, some causes should be provided for selecting these given concentrations rather than other concentrations.

4. In the section of both "1. Introduction" and "3. Results and discussion", some references are missing. These references are recommended as: Colloids and Surfaces A, 2021, 626 Article 127048; J. Mol. Liq., 2020, 316 Article 113793; J. Mol. Liq., 2019, 278: 53-60; J. Mol. Liq., 2018, 272: 380-386; J. Chem. Eng. Data, 2017, 62 (6), 1782-1787; J. Ind. Eng. Chem., 2016, 36, 263-270; Ind. Eng. Chem. Res. 2015, 54, 9683-9688; J. Ind. Eng. Chem., 2015, 30, 44-49; etc.
5. In the section of 2.2 Methods, it should be in detail stated to how to maintain the given temperature.
6. The conclusion should be improved.

Decision letter (RSOS-211204.R0)

Dear Dr Kumar:

Manuscript ID: RSOS-211204

Title: "Effects of various media on micellization, adsorption, and thermodynamic behavior of imipramine hydrochloride and antimicrobial surfactant mixtures"

Thank you for submitting the above manuscript to Royal Society Open Science. Your paper was sent to reviewers and their comments are included at the bottom of this letter.

In view of the concerns raised by the reviewers, the manuscript has been rejected in its current form. However, a new manuscript may be submitted which takes into consideration these comments.

Please note that resubmitting your manuscript does not guarantee eventual acceptance, and that your resubmission will be subject to peer review before a decision is made.

Your resubmitted manuscript should be submitted by 01-Mar-2022. If you are unable to submit by this date please contact the Editorial Office.

Yours sincerely,
Dr Ellis Wilde
Publishing Editor, Journals

Royal Society of Chemistry
Thomas Graham House
Science Park, Milton Road

Cambridge, CB4 0WF
 Royal Society Open Science - Chemistry Editorial Office

On behalf of the Subject Editor Professor Anthony Stace and the Associate Editor Dr Ya-Wen Wang

REVIEWER(S) REPORTS:

Associate Editor Comments to Author ():

RSC Associate Editor

Comments to the Author:

(There are no comments.)

RSC Subject Editor

Comments to the Author:

(There are no comments.)

Reviewers' Comments to Author:

Reviewer: 1

Comments to the Author(s)

This i'm afraid a poor paper, which is a pity as it is a potentially interesting subject. It is poorly written and contains a lot of background and explanation which to an informed reader should not be necessary. It contains the type of information that a thesis may contain and reads like a poor precis from someone's thesis. In general it come across as naive with an unawareness of the current state of the art in this area..

The data is limited and poorly presented, why, for example, are the variations in cmc for the mixtures not plotted. There seems to be no error bars in the data or the derived parameters, and indeed to quote many of those parameters to 2 decimal places is simply unrealistic. The thermodynamic analysis using RST but with an interaction parameters which varies with composition is simply not valid. The authors seem to be woefully unaware of recent applications of the pseudo phase approximation in which the asymmetry in the mixing, outside RST, is properly accounted for by expanding the excess free energy of mixing to higher order terms. Also the authors seem unaware of the limitations of the primarily surface tension approach and that the state of the art has now moved on considerably with the inclusions of new experimental approaches.

Reviewer: 2

Comments to the Author(s)

Authors focus on the interaction behavior between imipramine hydrochloride and benzethonium chloride in different media using different techniques. This work is interesting. The paper can be recommended to be published in this Journal after minor revisions are made as follows.

1. The English language in the whole manuscript should be polished.
2. The abstract should be improved.
3. In introduction, authors should give some explanations to select these materials (NaCl, urea, and thiourea) in this investigation. Also, some causes should be provided for selecting these given concentrations rather than other concentrations.
4. In the section of both "1. Introduction" and "3. Results and discussion", some references are missing. These references are recommended as: Colloids and Surfaces A, 2021, 626 Article 127048; J. Mol. Liq., 2020, 316 Article 113793; J. Mol. Liq., 2019, 278: 53-60; J. Mol. Liq., 2018, 272: 380-386; J. Chem. Eng. Data, 2017, 62 (6), 1782-1787; J. Ind. Eng. Chem., 2016, 36, 263-270; Ind. Eng. Chem.

Res. 2015, 54, 9683-9688; J. Ind. Eng. Chem., 2015, 30, 44-49; etc.

5. In the section of 2.2 Methods, it should be in detail stated to how to maintain the given temperature.

6. The conclusion should be improved.

Author's Response to Decision Letter for (RSOS-211204.R0)

See Appendix A.

RSOS-211527.R0

Review form: Reviewer 1

Is the manuscript scientifically sound in its present form?

No

Are the interpretations and conclusions justified by the results?

No

Is the language acceptable?

No

Do you have any ethical concerns with this paper?

No

Have you any concerns about statistical analyses in this paper?

No

Recommendation?

Reject

Comments to the Author(s)

The authors have made only a cursory attempt to address the serious shortcoming that i raised in the initial review. They have not by any means address the points satisfactorily. The paper remains poor and unacceptable, amd i can only recommend rejection

Review form: Reviewer 2

Is the manuscript scientifically sound in its present form?

Yes

Are the interpretations and conclusions justified by the results?

Yes

Is the language acceptable?

Yes

Do you have any ethical concerns with this paper?

No

Have you any concerns about statistical analyses in this paper?

No

Recommendation?

Accept as is

Comments to the Author(s)

Ok.

Review form: Reviewer 3

Is the manuscript scientifically sound in its present form?

Yes

Are the interpretations and conclusions justified by the results?

Yes

Is the language acceptable?

Yes

Do you have any ethical concerns with this paper?

No

Have you any concerns about statistical analyses in this paper?

No

Recommendation?

Accept with minor revision (please list in comments)

Comments to the Author(s)

Effect of various media on micellization and adsorption activity of varied mixtures of imipramine hydrochloride (IMP) drug and the antimicrobial cationic surfactant benzethonium chloride (BZCl) is investigated via a tensiometric method. In aqueous system, the interactions between component are also evaluated using UV-Visible and FTIR spectroscopy. Manuscript is showing good in current stage and can be accepted for publication after minor revision.

1. English is good throughout but still need some more polish.
2. Last sentence of abstract should be removed and given them in conclusion section only. No need here.
3. The sentence above equation (22): ...free energy change... give the symbol in bracket. In addition, compare between the value of binding free energy change and micellization Gibbs free energy.
4. In equation (8), author should give in nm². Authors should also change their value in table also.
5. Entirely the references should be checked and remove the errors, accordingly.

Decision letter (RSOS-211527.R0)

Dear Dr Kumar:

Title: Effects of various media on micellization, adsorption, and thermodynamic behavior of imipramine hydrochloride and antimicrobial surfactant mixtures
Manuscript ID: RSOS-211527

Thank you for submitting the above manuscript to Royal Society Open Science. On behalf of the Editors and the Royal Society of Chemistry, I am pleased to inform you that your manuscript will be accepted for publication in Royal Society Open Science subject to minor revision in accordance with the referee suggestions. Please find the reviewers' comments at the end of this email.

The reviewers and handling editors have recommended publication, but also suggest some minor revisions to your manuscript. Therefore, I invite you to respond to the comments and revise your manuscript.

Please also include the following statements alongside the other end statements. As we cannot publish your manuscript without these end statements included, if you feel that a given heading is not relevant to your paper, please nevertheless include the heading and explicitly state that it is not relevant to your work. We have included a screenshot example of the end statements for reference.

- Ethics statement

Please clarify whether you received ethical approval from a local ethics committee to carry out your study. If so please include details of this, including the name of the committee that gave consent in a Research Ethics section after your main text. Please also clarify whether you received informed consent for the participants to participate in the study and state this in your Research Ethics section.

OR

Please clarify whether you obtained the necessary licences and approvals from your institutional animal ethics committee before conducting your research. Please provide details of these licences and approvals in an Animal Ethics section after your main text.

OR

Please clarify whether you obtained the appropriate permissions and licences to conduct the fieldwork detailed in your study. Please provide details of these in your methods section.

- Data accessibility

It is a condition of publication that you make available the data and research materials supporting the results in the article. Datasets should be deposited in an appropriate publicly available repository and details of the associated accession number, link or DOI to the datasets must be included in the Data Accessibility section of the article (<https://royalsocietypublishing.org/rsos/for-authors#question17>). Reference(s) to datasets should also be included in the reference list of the article with DOIs (where available).

Please include a Data Availability section after your main text stating where supporting data are available from, or where they will be made available should your article be accepted for publication.

If you wish to submit your supporting data or code to Dryad (<http://datadryad.org/>), or modify your current submission to dryad, please use the following link:
<http://datadryad.org/submit?journalID=RSOS&manu=RSOS-211527>

- **Competing interests**

Please include a Competing Interests section after your main text declaring any financial or non-financial competing interests. If you have no competing interests please state 'I/we have no competing interests.

- **Authors' contributions**

Please include an Authors' Contributions section at the end of your main text detailing the contribution of each author. All authors should have read and approved the manuscript before submission and this should be stated in the Authors' Contributions section.

The list of Authors should meet all of the following criteria; 1) substantial contributions to conception and design, or acquisition of data, or analysis and interpretation of data; 2) drafting the article or revising it critically for important intellectual content; and 3) final approval of the version to be published.

- **Acknowledgements**

- **Funding statement**

Please include a funding section after your main text which lists the source of funding for each author.

Because the schedule for publication is very tight, it is a condition of publication that you submit the revised version of your manuscript before 13-Nov-2021. Please note that the revision deadline will expire at 00.00am on this date. If you do not think you will be able to meet this date please let me know immediately.

When submitting your revised manuscript, you will be able to respond to the comments made by the referees and upload a file "Response to Referees" in "Section 6 - File Upload". You can use this to document any changes you make to the original manuscript. In order to expedite the

processing of the revised manuscript, please be as specific as possible in your response to the referees.

Kind regards,
Dr Ellis Wilde
Publishing Editor, Journals

RSC Associate Editor
Comments to the Author:
According to the comments of the adjudicator, the decision was made.

RSC Subject Editor
Comments to the Author:
(There are no comments.)

Reviewer comments to Author:

Reviewer: 2

Comments to the Author(s)

Ok.

Reviewer: 1

Comments to the Author(s)

The authors have made only a cursory attempt to address the serious shortcoming that i raised in the initial review. They have not by any means address the points satisfactorily. The paper remains poor and unacceptable, amd i can only recommend rejection

Reviewer: 3

Comments to the Author(s)

Effect of various media on micellization and adsorption activity of varied mixtures of imipramine hydrochloride (IMP) drug and the antimicrobial cationic surfactant benzethonium chloride (BZCl) is investigated via a tensiometric method. In aqueous system, the interactions between component are also evaluated using UV-Visible and FTIR spectroscopy. Manuscript is showing good in current stage and can be accepted for publication after minor revision.

1. English is good throughout but still need some more polish.
2. Last sentence of abstract should be removed and given them in conclusion section only. No need here.
3. The sentence above equation (22): ...free energy change... give the symbol in bracket. In addition, compare between the value of binding free energy change and micellization Gibbs free energy.
4. In equation (8), author should give in nm². Authors should also change their value in table also.
5. Entirely the references should be checked and remove the errors, accordingly.

Author's Response to Decision Letter for (RSOS-211527.R0)

See Appendix B.

Decision letter (RSOS-211527.R1)

Dear Dr Kumar:

Title: Effects of various media on micellization, adsorption, and thermodynamic behavior of imipramine hydrochloride and antimicrobial surfactant mixtures
Manuscript ID: RSOS-211527.R1

It is a pleasure to accept your manuscript in its current form for publication in Royal Society Open Science. The chemistry content of Royal Society Open Science is published in collaboration with the Royal Society of Chemistry.

Yours sincerely,
Dr Ellis Wilde
Publishing Editor, Journals

RSC Associate Editor
Comments to the Author:
(There are no comments.)

Appendix A

Journal Title: Royal Society Open Science

Manuscript ID: RSOS-211204

Manuscript Title: Effects of various media on micellization, adsorption, and thermodynamic behavior of imipramine hydrochloride and antimicrobial surfactant mixtures

Dear Professor Ellis Wilde,

Thank you for your useful comments and suggestions on the structure of our manuscript. We have modified the manuscript accordingly, and the detailed corrections are listed below point by point.

Thanking you,

Sincerely yours'

Dr. Dileep Kumar

Response to Reviewer 1:

1. This i'm afraid a poor paper, which is a pity as it is a potentially interesting subject. It is poorly written and contains a lot of background and explanation which to an informed reader should not be necessary. It contains the type of information that a thesis may contain and reads like a poor precis from someone's thesis. In general it come across as naive with an unawareness of the current state of the art in this area. The data is limited and poorly presented, why, for example, are the variations in cmc for the mixtures not plotted. There seems to be no error bars in the data or the derived parameters, and indeed to quote many of those parameters to 2 decimal places is simply unrealistic. The thermodynamic analysis using RST but with an interaction parameters which varies with composition is simply not valid. The authors seem to be woefully unaware of recent applications of the pseudo phase approximation in which the asymmetry in the mixing, outside RST, is properly accounted for by expanding the excess free energy of mixing to higher order terms. Also the authors seem unaware of the limitations of the primarily surface tension approach and that the state of the art has now moved on considerably with the inclusions of new experimental approaches.

- ✓ Thanks. We tried our level best to make the manuscript worthy of the journal by removing grammatical and typing mistakes. Still, if any, kindly feel free to remove them.
- ✓ Thanks. Now the variations in cmc/cmc^{id} for the mixtures with change in mole fraction is plotted (see figure 2).
- ✓ Thanks. Now the error bars in figure 2 is given and uncertainty limits of derived parameters of the current study are given as footnote (see tables 2-4).
- ✓ Herein, it is found that value of interaction parameters was not constant through the change in mole fraction of ingredients which resources that the involved excess free energy will not be symmetrical, or it can say that molecular interactions diverge from symmetrical solution displaying the limitations of the Rubingh's model [a].
- ✓ It is clear from Table 2 that the values of β^m for the different mixtures was found to be not constant with the variation in the mole fraction of the bile salts. The variation of β^m values with composition is rather large, owing to the relative inaccuracy of the cmc determinations. The non-constancy of β^m with mixture composition showing the shortcomings of the Rubingh's approach for binary mixtures. For mixtures of an ionic amphiphililes, it has been disputed that the interaction parameter must be a function of the composition because the electrostatic contribution to β^m varies much with composition. Large differences in amphiphile head group size can also result in a composition dependent interaction parameter [b]. The non-consideration of the effects like counter-ion binding, chain length mismatch, ionic strength variation, etc. could be the reasons for affecting the evaluation procedure and the results. However, the quality of our data does not warrant analysis based on a composition dependent value of β^m . According to regular solution theory (RST), for a particular system β^m should remain independent of composition which is often not realized in practice. In this study, the parameter was found to be composition dependent like earlier reports on anionic–nonionic mixtures [c-f].

- (a) Reif, I.; Somasundaran, P. Asymmetric excess free energies and variable interaction parameters in mixed micellization. *Langmuir* **1999**, *15*, 3411-3417.
- (b) Eads, C. D.; Robosky, L. C. NMR studies of binary surfactant mixture thermodynamics: molecular size model for asymmetric activity coefficients. *Langmuir* **1999**, *15*, 2661-2668.
- (c) Ray, G. B.; Chakraborty, I.; Ghosh, S.; Moulik, S. P. On mixed binary surfactant systems comprising MEGA 10 and alkyltrimethylammonium bromides: A detailed physicochemical study with a critical analysis. *J. Colloid Interf. Sci.* **2007**, *307*, 543–553.
- (d) Jana, P. K.; Moulik, S. P. Interaction of bile salts with hexadecyltrimethylammonium bromide and sodium dodecyl sulfate. *J. Phys. Chem.* **1991**, *95*, 9525–9532.
- (e) Sharma, K. S.; Patil, S. R.; Rakshit, A. K.; Glenn, K.; Doiron, M.; Palepu, R. M.; Hassan, P. A. Self-aggregation of a cationic-nonionic surfactant mixture in aqueous media: tensiometric, conductometric, density, light scattering, potentiometric, and fluorometric studies. *J. Phys. Chem. B* **2004**, *108*, 12804-12812.
- (f) Rodríguez, A.; Graciani, M. del M.; Moreno-Vargas, A. J.; Moya, M. L. mixtures of monomeric and dimeric surfactants: hydrophobic chain length and spacer group length effects on non ideality. *J. Phys. Chem. B* **2008**, *112*, 11942–11949.

- ✓ Thanks. I am agreeing with your opinion that thermodynamic analysis using RST is not valid. There are various possible interactions between the entities of the solutions so complete account of the thermodynamic parameters associate is not possible as various factors like, charges, polarity, hydrophobicity, etc., are associated with them and due to these factors, the uncertainties in values are large. Because the technique used in the present study, it is not possible to consider the effect of various factors like charges, polarity, hydrophobicity, etc., associated with them to evaluate thermodynamic parameters. In the literature there are many papers have been published by using these equations to evaluate thermodynamic parameters (Thermochimica Acta 428 (2005) 147–

155; J. Colloid Interface Sci. 305 (2007) 293–300; J. Surfact. Deterg. 11 (2008) 287–292; J. Chem. Eng. Data 54 (2009) 559–565; J. Molecular Liquids 157 (2010) 113–118; J. Chem. Eng. Data 55 (2010) 4775–4779; J. Chem. Thermodynamics 43 (2011) 1349–1354; Colloids and Surfaces B 92 (2012) 203– 208; Journal of Molecular Liquids 222 (2016) 67–76; Journal of Molecular Liquids 277 (2019) 349–359; Journal of Molecular Liquids 322 (2021) 114558).

Response to Reviewer 2:

Thanks for recommending publication.

1. The English language in the whole manuscript should be polished.

✓ Thanks. We tried our level best to make the manuscript worthy of the journal by removing grammatical and typing mistakes. Still, if any, kindly feel free to remove them.

2. The abstract should be improved.

✓ Thanks. Now we have improved the abstract section of revised manuscript by rewriting.

3. In introduction, authors should give some explanations to select these materials (NaCl, urea, and thiourea) in this investigation. Also, some causes should be provided for selecting these given concentrations rather than other concentrations.

✓ Thanks for the useful comments. There was no specific reason for choosing aqueous, 50 mmol·kg⁻¹ NaCl, 300 mmol·kg⁻¹ urea (U) and 300 mmol·kg⁻¹ thiourea (TU) except to examine the effect of these additives that is usually found in the human body. Our main aim had been to show as to how the two constituents interact in aqueous solution as well as in the electrolyte or ureas solutions with the viewpoint of providing more information (thermodynamic and other) for the important and oft - used drug-surfactant combinations in the absence and presence of NaCl and urea in drug-delivery. The outcomes of electrolyte and ureas can give more knowledge for drug and surfactant mixtures

for developing improved delivery systems than aqueous system since the presence of electrolyte or ureas, the value of *cmc* of singular and mixture of amphiphiles declines or rises along with the spontaneity of solution mixture declines or rises. Detail information is forever valuable in attaining drugs' biological activity having the minimum unwanted effects (some explanation is given in revised manuscript of introduction section).

4. In the section of both “1. Introduction” and “3. Results and discussion”, some references are missing. These references are recommended as: Colloids and Surfaces A, 2021, 626 Article 127048; J. Mol. Liq., 2020, 316 Article 113793; J. Mol. Liq., 2019, 278: 53-60; J. Mol. Liq., 2018, 272: 380-386; J. Chem. Eng. Data, 2017, 62 (6), 1782–1787; J. Ind. Eng. Chem., 2016, 36, 263-270; Ind. Eng. Chem. Res. 2015, 54, 9683–9688; J. Ind. Eng. Chem., 2015, 30, 44-49; etc.

✓ Thanks. Suggested references are now included in revised manuscript (see Refs. # 4,9,13-16,20,56).

5. In the section of 2.2 Methods, it should be in detail stated to how to maintain the given temperature.

✓ Thanks provided in revised manuscript at relevant place.

6. The conclusion should be improved.

✓ Thanks for useful comments throughout the manuscript. Now conclusion section is improved by rewriting some section.

Appendix B

Journal Title: Royal Society Open Science

Manuscript Title: Effects of various media on micellization, adsorption, and thermodynamic behavior of imipramine hydrochloride and antimicrobial surfactant mixtures

Manuscript ID: RSOS-211527

Dear Professor Ellis Wilde,

Thank you for your useful comments and suggestions on the structure of our manuscript. We have modified the manuscript accordingly, and the detailed corrections are listed below point by point

Thanking you,
Dr. Dileep Kumar

Response to Reviewer: 2

1. Ok.

✓ Thanks for recommending publication.

Response to Reviewer: 1

Comments to the Author(s)

The authors have made only a cursory attempt to address the serious shortcoming that i raised in the initial review. They have not by any means address the points satisfactorily. The paper remains poor and unacceptable, amd i can only recommend rejection

✓ Thanks. We tried our level best by improving the manuscript according to comments raised by reviewer.

Response to Reviewer: 3

1. English is good throughout but still need some more polish.

✓ Thanks. We tried our level best to make the manuscript worthy of the journal by removing grammatical and typing mistakes. Still, if any, kindly feel free to remove them.

2. Last sentence of abstract should be removed and given them in conclusion section only. No need here.

✓ Thanks. Corrected as suggested by reviewer.

3. The sentence above equation (22): ...free energy change... give the symbol in bracket. In addition, compare between the value of binding free energy change and micellization Gibbs free energy.

✓ Thanks for the useful comments. Corrected as suggested and comparison of binding free energy change and micellization Gibbs free energy was included in the revised manuscript (see pages # 21 and 36).

4. In equation (8), author should give in nm². Authors should also change their value in table also.

✓ Thanks. Corrected as suggested throughout the manuscript. (see pages # 11 and 36).

5. Entirely the references should be checked and remove the errors, accordingly.

✓ Thanks. I have checked references section and corrected all errors throughout the reference.